# Theoretical Benefit and Limitation of Diffusion Language Model

**Guhao Feng**[1*]     **Yihan Geng**[2*]     **Jian Guan**[3]
**Wei Wu**[3]     **Liwei Wang**[1,4]     **Di He**[1†]

[1] State Key Laboratory of General Artificial Intelligence, Peking University
[2] School of Mathematical Sciences, Peking University
[3] Ant Group
[4] Center for Machine Learning Research, Peking University
{fenguhao,gengyihan}@stu.pku.edu.cn
{jianguanthu,wuwei19850318}@gmail.com
{wanglw,dihe}@pku.edu.cn

## Abstract

Diffusion language models have emerged as a new approach for text generation. By enabling the parallel sampling of multiple tokens in each diffusion step, they appear to offer a more efficient alternative to auto-regressive models. However, our observations show that current open-sourced diffusion language models require more sampling steps to achieve comparable accuracy on representative tasks–resulting in even higher inference costs than their auto-regressive counterparts. To investigate whether this is an inherent limitation, we conduct a rigorous theoretical analysis of a widely adopted variant: the Masked Diffusion Model (MDM). Surprisingly, our analysis reveals that the conclusion is highly sensitive to the choice of evaluation metric. Under mild conditions, we prove that when the target is near-optimal perplexity, MDMs can achieve this goal in a constant number of sampling steps, independent of sequence length. This result demonstrates that efficiency can, in principle, be attained without compromising generation quality. However, when targeting low sequence error rate–which is important for assessing the "correctness" of a generated sequence, such as a reasoning chain– we show that in the worst case, the required sampling steps must scale linearly with sequence length, thereby eliminating the efficiency advantage. Our analysis establishes the first theoretical foundation for understanding the comparative strengths and limitations of MDMs, offering practical guidance on when to favor MDMs over auto-regressive models and vice versa.

## 1 Introduction

Diffusion models (Ho et al., 2020; Song et al., 2021b) have emerged as a powerful paradigm in generative modeling, establishing state-of-the-art performance in image synthesis (Karras et al., 2022; Song et al., 2021a). Their extension to discrete domains has opened new possibilities for generating sequences, such as natural language (Campbell et al., 2022; Dieleman et al., 2022; Zheng et al., 2023; Lou et al., 2024; Campbell et al., 2024; Lovelace et al., 2024) and biological sequences (Rastogi et al., 2022; Vignac et al., 2022; Sun & Yang, 2023; Avdeyev et al., 2023). Among various discrete diffusion architectures, masked diffusion models (MDMs), which generate sequences by

---

*Equal Contribution

†Corresponding Author

39th Conference on Neural Information Processing Systems (NeurIPS 2025).

iteratively converting masks to tokens, have demonstrated competitive performance across language generation tasks (Shi et al., 2024; Sahoo et al., 2024; Ou et al., 2024).

While auto-regressive models generate sequences token-by-token, discrete diffusion models can generate multiple tokens simultaneously during each step, offering the potential for greater efficiency. However, efficiency and quality are often two sides of the same coin, and the key lies in identifying the trade-off points of different approaches. Unfortunately, as shown in Figure 1, we observed that for two recent open-sourced large MDMs, achieving performance comparable to that of left-to-right generative models incurs higher computational costs, as more sampling steps are required. This leads us to ask: do discrete diffusion models really offer a better trade-off than auto-regressive models, achieving superior efficiency while maintaining high-quality generated content? The answer may vary. If MDMs require fewer steps (i.e., neural network executions) without compromising quality, they could offer a more favorable trade-off than auto-regressive models. However, if the number of executions needed to maintain quality is similar to or exceeds that of auto-regressive models, then MDMs may not present a clear advantage.

To address the above question, we present the first theoretical analysis of the efficiency of Masked Diffusion Models (MDMs) and find that the efficiency-accuracy trade-off is highly sensitive to the choice of evaluation metric. We adopt two complementary metrics to assess the efficiency of MDMs in language modeling. The first metric, *token error rate* (TER), quantifies token-level accuracy, which correlates with the fluency of the generated text. In practice, *perplexity* is a widely used metric for measuring token-level errors of language models (Jelinek et al., 1977; Devlin et al., 2019); thus, we define the metric of TER by perplexity in this paper. The second metric, *sequence error rate* (SER), evaluates the correctness of an entire sequence, which is crucial for reasoning tasks requiring logically correct sequences. We provide a natural definition of SER that reflects the correctness of the whole sequence. Together, these metrics enable a comprehensive evaluation of MDMs at both the token and sequence levels.

We first provide a positive theoretical result regarding TER. We prove that under mild conditions, MDMs can achieve near-optimal TER with sampling steps regardless of the sequence length $L$. Compared to the auto-regressive model, which must be executed $L$ times to generate the sequence, MDMs demonstrate substantial efficiency gains, especially when the generation length is long. However, we show that this efficiency advantage diminishes when SER is considered. We theoretically prove that to achieve a low SER, in the worst case, the number of required sampling steps for MDMs must scale at least linearly with sequence length. Intuitively, this limitation arises from the fact that SER, as a metric for the entire sequence, requires the generated sequence to be free of any error in the whole sequence, which forces MDMs to sample only a small number of tokens per step to mitigate such inconsistencies. As a result, the number of required sampling steps can be significant. It is notable that each MDM sampling step usually incurs a higher computational cost than an auto-regressive step under the same architecture, thus MDMs offer no efficiency advantage.

To fully validate our theoretical findings, we conduct synthetic experiments and examine MDMs trained on formal languages, including $n$-gram languages and Hidden Markov Models (HMMs), systematically analyzing the relationship between performance and efficiency under both TER and SER metrics. All empirical results align with our theoretical predictions: For MDMs, achieving a low SER requires a significant number of sampling steps, and this requirement increases as sequence length grows. In contrast, obtaining a satisfactory TER demands fewer sampling steps, with this number remaining relatively constant regardless of sequence length. These findings offer practical guidance for selecting when to deploy diffusion language models based on specific application needs and requirements.

## 2 Related Work

**Discrete Diffusion Models.** The auto-regressive paradigm has achieved significant success in language modeling (Dai, 2019; Floridi & Chiriatti, 2020; Achiam et al., 2023). However, its left-to-right, token-by-token generation approach is not without limitations. Notably, it faces challenges such as restricted controllability (Zhang et al., 2023) and inefficiencies in inference speed (Leviathan et al., 2023). To overcome these drawbacks, inspired by the success of diffusion models in image generation (Sohl-Dickstein et al., 2015; Song et al., 2021a; Karras et al., 2022) researchers have adapted these techniques for NLP tasks (Austin et al., 2021a; He et al., 2022; Chen et al., 2022;

Meng et al., 2022; Ye et al., 2023; Gulrajani & Hashimoto, 2023; Zhang et al., 2024). Discrete diffusion models, in particular, have shown promising results, achieving comparable performance with auto-regressive models across a range of NLP benchmarks.

Discrete diffusion models can be categorized based on the initialization strategy of the reverse process: (1) reverse processes that begin with masked sequences and (2) reverse processes that start with sequences of tokens sampled randomly from the vocabulary. The first category, termed *masked diffusion models* (MDMs), includes models such as SEDD Absorb (Lou et al., 2024) and its streamlined variants in subsequent works (Sahoo et al., 2024; Zhao et al., 2024; Shi et al., 2024; Ou et al., 2024; Zheng et al., 2024). The second category encompasses models like SEDD Uniform (Lou et al., 2024), as well as extensions introduced in follow-up studies (Campbell et al., 2024). Notably, Gat et al. (2024); Davis et al. (2024) and Campbell et al. (2024) further extend flow-matching to the discrete domain, with differing initialization strategies: the former employs masked sequences, while the latter utilizes a customized distribution for the reverse process.

**Masked Diffusion Models.** Among the two primary classes of discrete diffusion models, MDMs have consistently demonstrated superior performance and scalability (Lou et al., 2024; Campbell et al., 2024). For instance, in Lou et al. (2024), the masked variant of SEDD significantly outperforms its uniform counterpart across a range of benchmarks. Similarly, Campbell et al. (2024) report that the masked variant achieves better results in most language tasks. Based on MDMs, some sampling strategies have been proposed to enhance efficiency or generation quality (Sahoo et al., 2024; Ou et al., 2024; Wang et al., 2025; Kim et al., 2025). Furthermore, recent advancements have successfully scaled MDMs to 8 billion parameters (Gat et al., 2024; Nie et al., 2024; Gong et al., 2024; Shi et al., 2024; Nie et al., 2025; Ye et al., 2025), underscoring their robustness and adaptability to large-scale NLP models. In this paper, we focus on MDMs, and our theoretical contributions can be applied to all MDMs, including the masked variant of discrete flow matching. Notably, concurrent with our theoretical analysis of MDMs, Li et al. (2024) also conduct an in-depth theoretical and empirical study on another class of MDMs. While their work primarily investigates the statistical complexity of learning these models, our analysis concentrates on the efficiency-accuracy trade-off of MDMs during inference.

## 3 Masked Diffusion Language Model

Without loss of generality, we study the sequence generation task where the sequence length is upper bounded by $L$. Let $\mathcal{V}$ denote the vocabulary. The MDM (Lou et al., 2024; Shi et al., 2024; Gong et al., 2024; Sahoo et al., 2024) extends the vocabulary $\mathcal{V}$ by introducing a special mask token $[\mathtt{m}]$. The forward diffusion process progressively transforms an initial sequence $\boldsymbol{x}_0 = (x_0^1, x_0^2, \ldots, x_0^L) \in \mathcal{V}^L$ into a fully masked sequence $\boldsymbol{x}_1 = ([\mathtt{m}], [\mathtt{m}], \ldots, [\mathtt{m}])$ by independently masking each token according to a predefined schedule. Conversely, the reverse process defines a generative model that reconstructs a sequence by iteratively modifying a fully/partially masked sequence. Below, we formally define both the forward and reverse processes.

**Forward process.** Given a sequence $\boldsymbol{x}_0$ and a masking schedule $\alpha_t$, the distribution of the sequence $\boldsymbol{x}_t$ at time $t \in [0, 1]$ is expressed as:

$$q_{t|0}(\boldsymbol{x}_t|\boldsymbol{x}_0) = \prod_{i=0}^{L-1} q_{t|0}(x_t^i|x_0^i),$$

$$\text{where} \quad q_{t|0}(x_t^i|x_0^i) = \begin{cases} \alpha_t, & x_t^i = x_0^i, \\ 1 - \alpha_t, & x_t^i = [\mathtt{m}]. \end{cases} \tag{1}$$

The masking schedule $\alpha_t$ is designed such that $\alpha_0 = 1$, ensuring that the sequence remains unmasked at the start of the process. Similar to the continuous diffusion methods (Ho et al., 2020; Song et al., 2021a; Karras et al., 2022), we set $\alpha_1 = 0$ (or a value approaching zero), ensuring the sequence is fully masked at the end of the forward process.

**Reverse process.** The reverse process reconstructs a sequence from a masked version by reversing the forward dynamics. Given the sequence at time $t$ and the original sequence $\boldsymbol{x}_0$, the conditional distribution of the sequence at time $s < t$, is defined as:

$$q_{s|t,0}(x_s^i|\boldsymbol{x}_t, \boldsymbol{x}_0) = \frac{1 - \alpha_s}{1 - \alpha_t} \delta_{x_t^i}(x_s^i) + \frac{\alpha_s - \alpha_t}{1 - \alpha_t} \delta_{x_0^i}(x_s^i),$$

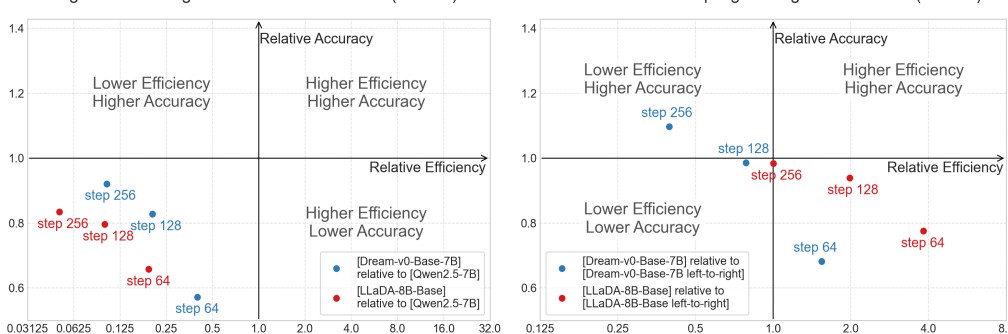

Figure 1: Efficiency and accuracy of MDMs on GSM8K (8-shot): The left figure uses Qwen2.5-7B as the evaluation baseline and plots the relative accuracy and efficiency of Dream-v0-Base-7B and LLaDA-8B-Base across different sampling steps. The origin can be considered as the baseline result. The first quadrant demonstrates models that can simultaneously achieve higher efficiency and accuracy. From the figure, we can see that the MDMs all fall into the third quadrant, indicating lower efficiency and lower performance. The right figure uses a baseline in which the MDMs are constrained to generate the left-most masked token at each step, effectively mimicking auto-regressive decoding using the same model. Still, even under this setting, none of the configurations fall into the first quadrant.

where $\delta_x(y)$ is the Kronecker delta function. Marginalizing over $\mathbf{x}_0$ yields the true reverse process $q(\mathbf{x}_s|\mathbf{x}_t)$:

$$q_{s|t}(\boldsymbol{x}_s|\boldsymbol{x}_t) = \prod_{i=0}^{L-1} q_{s|t}(x_s^i|\boldsymbol{x}_t), \quad \text{where}$$

$$q_{s|t}(x_s^i|\boldsymbol{x}_t) = \begin{cases} 1, & x_t^i \neq [\mathtt{m}], x_s^i = x_t^i, \\ \frac{1-\alpha_s}{1-\alpha_t}, & x_t^i = [\mathtt{m}], x_s^i = [\mathtt{m}], \\ \frac{\alpha_s-\alpha_t}{1-\alpha_t} q_{0|t}(x_s^i|\boldsymbol{x}_t), & x_t^i = [\mathtt{m}], x_s^i \neq [\mathtt{m}], \\ 0, & \text{otherwise.} \end{cases} \tag{2}$$

In MDM, a parameterized reverse model $p_\theta$ is often employed to approximate the distribution $q_{0|t}(x_s^i|\boldsymbol{x}_t)$. This model is trained by minimizing the evidence lower bound (ELBO) (Lou et al., 2024; Shi et al., 2024; Gong et al., 2024; Sahoo et al., 2024) on the negative log-likelihood of the data distribution $q_0$.

**Inference.** Inference within the MDM framework entails discretizing the reverse process to iteratively reconstruct sequences from a fully masked sequence. Let $T$ denote the number of sampling steps. Starting with a fully masked sequence, the denoising process proceeds via $q_{s|t}(\boldsymbol{x}_s \mid \boldsymbol{x}_t)$, where $s = \frac{i}{T}$ and $t = \frac{i+1}{T}$. At each step, the model first samples $\boldsymbol{x}_0$ from the conditional distribution $p_\theta(\boldsymbol{x}_0 \mid \boldsymbol{x}_t)$, followed by masking specific tokens according to $q(\boldsymbol{x}_s \mid \boldsymbol{x}_t, \boldsymbol{x}_0)$.

In practice, the reverse model is parameterized using a factorized denoising model, where the conditional distribution $p_\theta(\boldsymbol{x}_0 \mid \boldsymbol{x}_t)$ is expressed as:

$$p_\theta(\boldsymbol{x}_0 \mid \boldsymbol{x}_t) = \prod_{i=1}^{L} p_\theta(x_0^i \mid \boldsymbol{x}_t). \tag{3}$$

Here, each token is predicted independently using $p_\theta(x_0^i \mid \boldsymbol{x}_t)$, allowing for efficient parallel sampling. However, this factorized approach imposes a significant limitation: it disregards interdependencies between tokens within the sequence. As a result, the factorized model $p_\theta(\boldsymbol{x}_0 \mid \boldsymbol{x}_t)$ cannot exactly match the true reverse distribution $q(\boldsymbol{x}_0 \mid \boldsymbol{x}_t)$ (Xu et al., 2024). In this work, we analyze the conditions under which this sampling method achieves a favorable balance between efficiency and the quality of the generated sequences. Motivated by these observations, this paper begins by re-examining the theoretical upper bounds on the performance and efficiency of MDMs.

**Efficiency-accuracy trade-off of open-sourced MDMs.** The efficiency-accuracy trade-off is crucial for understanding the strengths of different generative models. In Lou et al. (2024); Nie et al.

(2024), researchers demonstrate that MDMs offer a more favorable trade-off than auto-regressive models in terms of perplexity-that is, the learned MDMs can achieve competitive perplexity with higher token throughput. However, to the best of our knowledge, there has been limited empirical investigation of this trade-off in realistic tasks such as reasoning.

To address this gap, we conducted experiments on the GSM8K, MBPP, and HumanEval benchmarks (Cobbe et al., 2021; Austin et al., 2021b; Chen et al., 2021), using two recently released large-scale MDMs in 2025, Dream-v0-Base-7B and LLaDA-8B-Base (Ye et al., 2025; Nie et al., 2025). We directly evaluate these checkpoints using different numbers of decoding steps, and compare the accuracy and efficiency against similar-size auto-regressive baseline Qwen2.5-7B (Team, 2024). The results on GSM8K are presented in Figure 1, the results on MBPP and HumanEval are presented in Appendix A.3 and Appendix A.4. Notably, across all configurations (sampling steps), the performance and efficiency of MDMs are significantly worse than that of auto-regressive models.

A key contributing factor to this gap is that the model checkpoints were trained on different private datasets, which inherently introduces bias into the comparison. To ensure fairness, we also construct a baseline by running the MDMs in an auto-regressive manner: at each diffusion step, the model is constrained to generate the left-most masked token, thereby producing the full sequence in $L$ steps. However, even compared to this naive baseline, none of the configurations fall into the first quadrant of the plot in Figure 1-indicating that, compared to auto-regressive decoding, MDMs may not be able to simultaneously achieve superior efficiency and improved performance. Full details are given in Appendix A.

# 4 Theoretical Analysis

Motivated by the observations above, this paper begins by re-examining the theoretical trade-off between the accuracy and efficiency of MDMs. In image generation, the primary goal is typically to produce visually appealing and seamless images (Heusel et al., 2017). Language generation is more task-specific. Depending on the application, the users may prefer fluent outputs, as in article writing, or precise and accurate reasoning, as in problem-solving tasks. In this section, we explore the sampling efficiency of MDMs in addressing various language tasks with respect to different evaluation metrics.

## 4.1 Notations and Problem Setting

Our investigation employs the hidden Markov model (HMM) framework to analyze natural language generation. This section establishes the formal notation and problem setting that underlie our subsequent analysis.

HMMs (Eddy, 1996) provide a probabilistic foundation for modeling sequential data with latent structures, where observed sequences are generated by an underlying sequence of unobservable hidden states. Formally, an HMM $\mathcal{H} = (\mathcal{S}, \mathcal{V}, \boldsymbol{A}, \boldsymbol{B})$ is characterized by the following components: a finite set of hidden states $\mathcal{S} = \{s_1, s_2, \ldots, s_N\}$, an observable vocabulary $\mathcal{V}$, a state transition probability matrix $\boldsymbol{A} \in \mathbb{R}^{N \times N}$, an emission probability matrix $\boldsymbol{B} \in \mathbb{R}^{N \times |\mathcal{V}|}$, and an initial state distribution $\boldsymbol{\pi} \in \mathbb{R}^N$. Given a sequence of observations $\boldsymbol{x} = (x_1, x_2, \ldots, x_L) \in \mathcal{V}^L$ and a sequence of hidden states $\boldsymbol{s} = (s_1, s_2, \ldots, s_L) \in \mathcal{S}^L$, the generative process of an HMM is governed by the following probabilistic relations:

$$\Pr(s_1) = \boldsymbol{\pi}_{s_1}, \quad \Pr(x_i \mid s_i) = \boldsymbol{B}_{s_i, x_i},$$
$$\Pr(s_i \mid s_{1:i-1}) = \Pr(s_i \mid s_{i-1}) = \boldsymbol{A}_{s_{i-1}, s_i}.$$

This formulation enables HMMs to capture both the sequential dependencies among hidden states and their probabilistic relationships with observed data. In the field of NLP, HMMs serve as the fundamental statistical tools to model natural language (Eddy, 1996; Marti & Bunke, 2001). A notable special case of HMM is the $n$-gram language model (Brown et al., 1992), which estimates the probability of a token given its preceding $n - 1$ tokens. Despite their simplicity, $n$-gram models are foundational tools in NLP tasks (Brown et al., 1992; De Novais et al., 2010). Moreover, Liu et al. (2024) suggests that scaling up $n$-gram models can also achieve performance comparable to modern large language models.

Formally, we aim to address the following question: If MDMs have the capability to approximate a target HMM model, what are the computational costs, and do MDMs offer advantages over auto-

regressive models? To evaluate the approximation quality of MDMs, we adopt two widely used metrics: *TER* and *SER*, which quantify different aspects of a model's performance.

**Token Error Rate.** In practice, perplexity is one of the most widely used metrics for evaluating token-level errors in language models. It quantifies the uncertainty of a model in predicting the next token in a sequence and serves as a standard measure for assessing the quality of text generation. In this paper, we define the TER by perplexity. Models with lower TER are generally considered more effective at generating fluent and coherent text. Formally, given a ground-truth language model $q$ and an evaluated model $p$, the TER is computed as:

$$\mathrm{TER}(p) = 2^{\mathbb{E}_{\boldsymbol{x} \sim q}\left[-\frac{\log(p(\boldsymbol{x}))}{|\boldsymbol{x}|}\right]}. \tag{4}$$

**Sequence Error Rate.** The SER evaluates the correctness of an entire sequence rather than individual tokens. Let $q$ represent a target language defined over a vocabulary $\mathcal{V}$, and let $\mathcal{L}_q = \{\boldsymbol{x} \in \mathcal{V}^* \mid q(\boldsymbol{x}) > 0\}$ denote the support set of distribution $q$. For a generative model $p$, the SER is defined as:

$$\mathrm{SER}(p) = 1 - \sum_{\boldsymbol{x} \in \mathcal{L}_q} p(\boldsymbol{x}). \tag{5}$$

This metric quantifies the probability that the model generates sequences falling outside the support set of the ground-truth distribution.

Compared to TER, SER imposes a stricter evaluation criterion by requiring the correctness of entire sequences. This makes SER particularly well-suited for tasks that demand logical consistency or reasoning, where the correctness of the complete reasoning chain is crucial.

## 4.2 MDMs Can Generate Low-TER Sentences Efficiently

In this subsection, we rigorously examine the efficiency of sampling in MDMs, demonstrating that MDMs are capable of efficiently generating sentences with near-optimal TER. To establish the main theoretical results, we assume that the MDMs have enough expressive power and begin with the following assumption:

**Assumption 4.1** (Learning with Small Error)**.** Let $q$ denote the target language model with vocabulary $\mathcal{V}$, and let $p_\theta$ represent the reverse model trained to approximate the reverse process generating the target language under a masking schedule $\alpha_t$. Assume there exists $\epsilon_{\text{learning}} > 0$ such that the KL divergence between $p_\theta$ and the reverse process distribution generating the language $q$ is bounded by $\epsilon_{\text{learning}}$, i.e.,

$$D_{\mathrm{KL}}(q_{0|t}(x_0^i \mid \boldsymbol{x}_t) \| p_\theta(x_0^i \mid \boldsymbol{x}_t)) < \epsilon_{\text{learning}}, \quad \forall\, t \text{ and } \boldsymbol{x}_t.$$

It is worth noting that $p_\theta(x_0^i \mid \boldsymbol{x}_t) = q_{0|t}(x_0^i \mid \boldsymbol{x}_t)$ represents the optimal solution to the ELBO loss during training. Assumption 4.1 implies that the MDM model is well-trained and approximates the ground-truth distribution with only a small error.

During MDM inference, the time interval $[0, 1]$ is discretized into $N$ steps, where $t_i = \frac{i}{N}$, $i \in [N]$, and iteratively reconstruct sequences from a fully masked sequence. The following theorem shows that the sequence distribution generated by the reverse process, even with a small number of sampling steps, can achieve near-optimal TER. Consequently, MDMs exhibit high efficiency in generating $n$-gram language.

**Theorem 4.2** (TER Bounds for $n$-Gram Language Generation)**.** *For any $n$-gram language $q$ and any $\epsilon > 0$, let $p_\theta$ denote the reverse model and $L$ denote the sequence length. The distribution over sequences generated by $p_\theta$ is denoted as $p$. For any $L > O\left(\frac{n-1}{\epsilon^{n+0.5}}\right)$, under Assumption 4.1, there exists a masking schedule $\alpha_t$ such that, with $N = O\left(\frac{n-1}{\epsilon^n}\right)$ sampling steps, the TER of the MDM is upper-bounded by:*

$$\log \mathrm{TER}(p) \le \log \mathrm{TER}(q) + \epsilon_{learning} + 4\epsilon \log |\mathcal{V}|. \tag{6}$$

The proof of this theorem is presented in Appendix D. Theorem 4.2 demonstrates that MDMs can efficiently generate sentences with high fidelity. It is notable that for a given data distribution $q$, the TER of a language model $p$ achieves its global minimum when $p = q$. To ensure a gap of at most $\epsilon$ with the optimal TER during sampling, the number of required sampling steps is bounded by $O\left(\frac{n-1}{\epsilon^n}\right)$.

The above results suggest that to achieve near-optimal TER, MDMs require only a number of sampling steps that is independent of the sequence length $L$. In each sampling step, the neural network

model, i.e., a Transformer, is executed once. Therefore, informally, the neural network execution count is constant for MDM. This offers substantial efficiency gains over auto-regressive models, where the model must be executed $L$ times, once for each token in the sequence. Such efficiency enables MDMs to handle long-sequence generation tasks effectively while maintaining high-quality outputs.

## 4.3 MDMs Cannot Generate Low-SER Sentences with A Low Cost

In this subsection, we examine the SER of sampling in MDMs and highlight a fundamental limitation of MDMs in generating logically rigorous language. We begin by establishing that, with sufficient sampling steps, the MDMs are able to approximate a target HMM model with perfect SER.

**Theorem 4.3** (Accurate Generation of HMM with Sufficient Steps). *Let $q$ denote any HMM, and let $p_\theta$ represent the reverse model under an arbitrary masking schedule, where $L$ is the sequence length. Let $p$ denote the distribution over sequences generated by $p_\theta$. Under Assumption 4.1 with a learning error $\epsilon_{learning} < O(\frac{\delta}{L})$, and given a sufficient number of reverse steps, the sequence error rate $\mathrm{SER}(p)$ of the generated text satisfies $\mathrm{SER}(p) \leq \delta$.*

The complete proof of Theorem 4.3 is detailed in Appendix E.1. While this result establishes the theoretical capability of MDMs to achieve low SER, we still need to estimate the computational cost to achieve it. The following theorem provides a negative result for this problem.

**Theorem 4.4** (SER Bound for HMM Generation). *There exists an HMM $q$ over a vocabulary of size 16 that satisfies the following conditions: for any reverse model $p_\theta$ under Assumption 4.1 with $\epsilon_{\mathrm{learning}} < \frac{1}{128}$, and any masking schedule $\alpha_t$, let $p$ denote the distribution over sequences generated by $p_\theta$. There exists a constant $C$ such that if the number of sampling steps satisfies $N = CL$, where $L$ is the sequence length, the SER of the generated text is lower-bounded by: $\mathrm{SER}(p) > \frac{1}{2}$.*

The proof of Theorem 4.4 is presented in Appendix E.2. Theorem 4.4 shows that to generate sequences with low SER, the number of sampling steps in MDMs must scale at least linearly with the sequence length $L$, indicating that the number of neural network executions is comparable between MDMs and autoregressive models. However, this scaling law of MDMs typically leads to much higher computational costs compared to autoregressive models. For instance, in the case of Transformer-based architectures, each execution step in MDMs involves a quadratic computational complexity in terms of $L$, as opposed to the linear complexity of auto-regressive Transformer models in each generation step (through reusing the stored KV caches). Consequently, in accuracy-critical applications, MDMs offer no computational efficiency advantage over auto-regressive models.

**Extended analysis of Theorem 4.4** Since the high SER partly arises from irreducible errors introduced by parallel sampling, it is natural to ask whether this issue can be mitigated by allowing tokens to be remasked. In Appendix E.3, we examine the remasking strategy recently proposed by (Wang et al., 2025), and demonstrate that Theorem 4.4 continues to hold. Furthermore, some prior works (Sahoo et al., 2024; Ou et al., 2024) have proposed efficient sampling strategies that reuse cached outputs without requiring additional forward passes through the network when no token is modified from [m] at a given step. Nevertheless, our theoretical results remain applicable to these sampling strategies, which is discussed in Appendix B.3.

**Discussion regarding theory and GSM8k result.** SER is closely related to the accuracy of MDMs in solving mathematical problems, as an incorrect chain of thoughts typically leads to an erroneous answer (Wei et al., 2022b; Yu et al., 2024; Zhu et al., 2024). Our theory reveals that MDMs incur higher computational costs when evaluating generations by SER. This finding is consistent with results presented in Figure 1. Both our theoretical insights and these prior experiments demonstrate that when MDMs are applied to math problems, their efficiency-performance trade-off is less favorable than that of AR models. For further discussion, please refer to Appendix B.2.

**Practical guideline of ARs v.s. MDMs.** Based on the theoretical results, conclusions regarding when to favor MDMs depend heavily on the evaluation metric employed. Specifically, MDMs excel in applications where fluency is prioritized. In contrast, for reasoning-intensive tasks that demand highly accurate trajectories, MDMs may fail to offer a significant efficiency advantage over auto-regressive models. As a result, MDMs are better suited for fluent generation tasks, while ARs remain the preferred choice for tasks that require precise, step-by-step reasoning.

| **SER with Different Sampling Steps** | | | | | | | |
|---|---|---|---|---|---|---|---|
| **N-gram** | **Length** | **128** | **256** | **512** | **1024** | **2048** | **AR** |
| **2-gram** | **512** | $36.2 \pm 4.2$ | $24.5 \pm 3.2$ | $14.4 \pm 2.9$ | $7.4 \pm 2.2$ | $5.4 \pm 1.2$ | $0.2 \pm 0.1$ |
| | **1024** | $59.5 \pm 4.9$ | $40.3 \pm 4.6$ | $23.2 \pm 3.2$ | $14.6 \pm 3.0$ | $9.4 \pm 1.5$ | $0.3 \pm 0.2$ |
| | **2048** | $88.70 \pm 2.8$ | $71.5 \pm 3.7$ | $50.2 \pm 4.4$ | $40.2 \pm 3.9$ | $26.9 \pm 3.3$ | $0.3 \pm 0.1$ |
| **3-gram** | **512** | $37.3 \pm 3.6$ | $21.6 \pm 3.3$ | $16.4 \pm 2.7$ | $11.6 \pm 3.0$ | $7.9 \pm 2.1$ | $0.2 \pm 0.1$ |
| | **1024** | $64.8 \pm 4.7$ | $39.4 \pm 5.3$ | $26.6 \pm 3.0$ | $19.3 \pm 3.2$ | $13.9 \pm 2.7$ | $0.3 \pm 0.3$ |
| | **2048** | $86.0 \pm 3.9$ | $66.1 \pm 5.3$ | $48.2 \pm 5.2$ | $48.2 \pm 5.6$ | $37.8 \pm 5.2$ | $0.2 \pm 0.2$ |
| **4-gram** | **512** | $51.8 \pm 4.4$ | $39.2 \pm 3.2$ | $34.0 \pm 3.5$ | $30.0 \pm 2.7$ | $25.7 \pm 2.6$ | $0.5 \pm 0.1$ |
| | **1024** | $67.9 \pm 5.6$ | $52.9 \pm 5.1$ | $41.8 \pm 2.7$ | $36.1 \pm 5.8$ | $35.4 \pm 5.1$ | $0.4 \pm 0.2$ |
| | **2048** | $80.0 \pm 4.1$ | $68.1 \pm 5.5$ | $62.0 \pm 5.4$ | $62.0 \pm 4.3$ | $60.1 \pm 3.8$ | $0.5 \pm 0.4$ |
| **HMM** | **512** | $19.1 \pm 2.1$ | $16.7 \pm 2.3$ | $12.8 \pm 2.6$ | $9.1 \pm 2.6$ | $8.2 \pm 2.2$ | $0.4 \pm 0.1$ |
| | **1024** | $40.1 \pm 2.5$ | $32.8 \pm 3.1$ | $29.9 \pm 2.6$ | $30.5 \pm 3.0$ | $28.0 \pm 3.3$ | $0.4 \pm 0.2$ |
| | **2048** | $68.5 \pm 5.6$ | $63.4 \pm 4.2$ | $62.9 \pm 5.5$ | $62.9 \pm 4.7$ | $61.8 \pm 5.8$ | $0.3 \pm 0.1$ |

| **TER with Different Sampling Steps** | | | | | | | |
|---|---|---|---|---|---|---|---|
| **N-gram** | **Length** | **128** | **256** | **512** | **1024** | **2048** | **AR** |
| **2-gram** | **512** | $3.71 \pm .03$ | $3.68 \pm .02$ | $3.67 \pm .01$ | $3.67 \pm .01$ | $3.67 \pm .01$ | $3.67 \pm .01$ |
| | **1024** | $3.72 \pm .05$ | $3.70 \pm .03$ | $3.68 \pm .01$ | $3.68 \pm .02$ | $3.67 \pm .01$ | $3.67 \pm .01$ |
| | **2048** | $3.71 \pm .05$ | $3.69 \pm .03$ | $3.67 \pm .02$ | $3.66 \pm .01$ | $3.66 \pm .01$ | $3.66 \pm .02$ |
| **3-gram** | **512** | $3.13 \pm .05$ | $3.10 \pm .03$ | $3.07 \pm .01$ | $3.07 \pm .01$ | $3.07 \pm .01$ | $3.07 \pm .01$ |
| | **1024** | $3.14 \pm .06$ | $3.10 \pm .03$ | $3.09 \pm .02$ | $3.08 \pm .01$ | $3.08 \pm .01$ | $3.08 \pm .02$ |
| | **2048** | $3.15 \pm .06$ | $3.11 \pm .03$ | $3.10 \pm .02$ | $3.09 \pm .01$ | $3.08 \pm .02$ | $3.08 \pm .02$ |
| **4-gram** | **512** | $3.27 \pm .08$ | $3.23 \pm .04$ | $3.20 \pm .02$ | $3.20 \pm .01$ | $3.19 \pm .01$ | $3.19 \pm .01$ |
| | **1024** | $3.21 \pm .08$ | $3.16 \pm .03$ | $3.16 \pm .02$ | $3.14 \pm .01$ | $3.14 \pm .01$ | $3.14 \pm .01$ |
| | **2048** | $3.20 \pm .06$ | $3.16 \pm .03$ | $3.14 \pm .02$ | $3.13 \pm .02$ | $3.12 \pm .01$ | $3.12 \pm .02$ |
| **HMM** | **512** | $4.04 \pm .03$ | $4.01 \pm .03$ | $4.00 \pm .01$ | $3.99 \pm .01$ | $3.99 \pm .01$ | $3.99 \pm .01$ |
| | **1024** | $4.03 \pm .04$ | $4.01 \pm .02$ | $4.00 \pm .01$ | $3.99 \pm .02$ | $3.98 \pm .02$ | $3.98 \pm .02$ |
| | **2048** | $4.03 \pm .02$ | $4.02 \pm .01$ | $4.01 \pm .01$ | $4.00 \pm .01$ | $4.00 \pm .01$ | $4.00 \pm .01$ |

Table 1: SER and TER results for MDMs (n-grams, $n \in \{2, 3, 4\}$; HMMs) vs. AR baseline across varying sequence lengths (512, 1024, 2048) and sampling steps. The 'AR' column (underlined) provides the baseline for comparison. In the TER table, blue highlighting indicates MDM performance statistically similar to the AR baseline according to T-Test. MDMs approach AR TER with $\approx 512$ sampling steps, irrespective of sequence length. Accuracy, however, requires more steps for longer sequences and generally lags behind AR models with a large margin.

## 5 Experiments

As discussed in the previous section, our theory aligns with observations in natural language tasks. However, for comprehensive validation and completeness, we also conducted experiments to empirically support our theoretical results. More precisely, these experiments were designed to investigate the relationship between sampling steps, sequence length, the SER, and the TER. We will first introduce our experimental setup, followed by a presentation of the results.

### 5.1 Experimental Setup

**Tasks and Datasets.** We evaluated MDMs on several formal languages: $n$-gram languages (with $n \in \{2, 3, 4\}$) and HMMs. For each language type, parameters (e.g., transition matrices,

observation matrices, initial distributions) were randomly sampled. A detailed description of this generation process and examples of resulting sequences are available in Appendix F.1. These formal languages were used to generate datasets of 1,000,000 samples each, with 990,000 for training and 10,000 for validation. Datasets were generated with sequence lengths $L \in \{512, 1024, 2048\}$.

**Model Training.** Transformer-based architectures served as the backbone for our MDMs, chosen for their scalability and expressiveness in sequence modeling. Detailed architectural specifications, including layer counts, hidden dimensions, and positional encoding schemes, are provided in Table 7 (Appendix F.2). The training procedure largely followed the framework of Sahoo et al. (2024), with specific training configurations detailed in Table 8. Models were trained for 20 epochs, with convergence monitored on the validation set using perplexity. The trained models successfully achieved perplexity values consistent with the ground-truth language models generating the datasets.

**Evaluation Metrics.** To evaluate the quality of generated sequences in line with our theoretical framework, we used TER and SER as primary metrics. Generative perplexity, computed using the ground-truth model, served as the TER metric. SER was calculated directly via Equation (5) (where accuracy reported in tables is $1 - \text{SER}$), also utilizing the ground-truth models. For sequence generation, we employed the `ddpm_cache` sampler from Sahoo et al. (2024); its influence under varying sampling steps is further discussed in Appendix B.3. We report the inference time per sequence of MDMs with 512 sampling steps and AR models in Table 2 across different sequence length (detailed inference settings in Appendix F.2). For robust evaluation, all reported TER and SER values are averages over 2000 generated sequences for each experimental setting.

**Auto-Regressive Baseline.** We also trained AR models with identical architectures and model sizes on the same datasets generated by the formal languages for comparison. These AR models were evaluated using the same metrics. Training configurations for the AR models are detailed in Table 9.

## 5.2 Experiment Results

The experiment results are presented in Section 5. This table presents the generative perplexity and accuracy on n-gram languages ($n \in \{2, 3, 4\}$) and HMM. The results are detailed for sequence lengths of 512, 1024, and 2048, across a varying number of sampling steps (128, 256, 512, 1024, and 2048). The final 'AR' column shows the performance of the auto-regressive baseline.

As Section 5 shows, under the metric of TER, MDMs achieve near-optimal generative perplexity, closely matching the AR baseline, with approximately 512 sampling steps. Crucially, this required number of sampling steps remains relatively constant across the tested sequence lengths (512, 1024, 2048). This empirically supports our theoretical finding that TER is primarily dependent on the number of sampling steps, rather than sequence length.

In contrast, achieving low SER with MDMs necessitates a substantially larger number of sampling steps. This requirement becomes more pronounced as sequence length increases. For instance, for 2-gram models generating sequences of length 2048, even 2048 sampling steps yield only 73.1% accuracy, whereas AR models achieve near 100%. This aligns with our theory that SER is more sensitive to both the number of sampling steps and the sequence length.

Furthermore, even with a high number of sampling steps (e.g., 2048), a significant gap in SER persists between MDMs and AR models. This is particularly evident for more complex languages like 4-grams and HMMs across all sequence lengths. For example, with 4-gram language of length 512, MDMs with 2048 steps achieve 74.3% accuracy, while the AR baseline reaches 99.5%. AR models, due to their token-by-token generation mechanism, consistently achieve near-perfect SER on these tasks. This result further validate our theoretical result that MDMs Cannot generate low-SER sentences with a low cost.

| Sequence Length | 512 | 1024 | 2048 |
|---|---|---|---|
| MDMs (512 steps) | 3.1s | 4.2s | 4.7s |
| AR | 1.7s | 3.3s | 7.0s |

Table 2: The inference time per sequence of MDMs with 512 sampling steps and AR models.

Despite the challenges in achieving low SER, MDMs demonstrate considerable efficiency in generating fluent sequences with a fixed number of sampling steps (e.g., 512 steps), especially for longer sequences. The inference time per sequence, detailed in Table 2, quantifies this advantage, showing MDMs to be highly efficient for generating fluent long sequences.

These empirical findings underscores the trade-off between generation efficiency and performace for MDMs. While MDMs excel at producing fluent outputs efficiently, they require considerably more sampling iterations to achieve low SER. This characteristic is particularly salient for reasoning-intensive tasks that demand high sequence-level correctness. Collectively, these experimental results provide strong empirical corroboration for our theoretical analyses.

# 6 Conclusion and Limitations

This paper provides a rigorous theoretical analysis of the efficiency of MDMs under various metrics. We demonstrate that MDMs can achieve near-optimal TER with a fixed number of sampling steps, regardless of sequence length, making them highly efficient for tasks emphasizing fluency. However, when evaluated using SER, MDMs require sampling steps that scale linearly with sequence length, negating their efficiency advantage over auto-regressive models. Our study focuses on formal languages modeled using HMM, which, while foundational, still differs from modern language models. Extending this analysis to more advanced language models remains an important direction for future work. Additionally, we primarily analyze Masked Diffusion Models, but a broader family of diffusion-based language models, including variants like SEDD-unform (Lou et al., 2024). Further exploration is needed to generalize our findings to real-world settings and to systematically analyze other diffusion approaches.

## Acknowledgement

This work is supported in part by the State Key Laboratory of General Artificial Intelligence. DH is supported by National Science Foundation of China (NSFC62376007) and Beijing Natural Science Foundation (Z250001).

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

# Appendix

## A  Details of Preliminary Experiments

### A.1  Models and Sampling Strategies Compared

We follow Ye et al. (2025) and Nie et al. (2025) to setup our experiments on GSM8K, MBPP and HumanEval benchmarks. For each benchmark, we compare the following sampling strategies of different models as suggested:

- **Entropy-based sampling** (Ye et al., 2025) applied to Dream-v0-Base-7B, with different sampling steps.
- **Low-confidence-based sampling** (Nie et al., 2025) applied to LLaDA-8B-Base, also with different sampling steps.
- **Left-to-right (AR-style) sampling strategy** for Dream-v0-Base-7B and LLaDA-8B-Base, where one token is generated per step in a sequential, left-to-right manner.
- **Standard auto-regressive generation** for Qwen2.5-7B with default settings.

Here, the entropy-based sampling strategy and the low-confidence-based sampling strategy are the recommended sampling strategies for the MDMs (Ye et al., 2025; Nie et al., 2025). Qwen2.5-7B serves as the standard AR baseline, while the left-to-right sampling strategy of MDMs serves as a naive baseline for the MDMs, aiming to eliminate the influence of other contributing factors, including different training datasets or training methods.

For Dream-v0-Base-7B, we manually implemented the left-to-right sampling process. In the case of LLaDA-8B-Base, setting `block_length=1` during generation naturally obtains the same behavior. This configuration does not stop generation after encountering `EOS`, which may cause an increased time cost, but our results indicate that this behavior does not affect the overall conclusions.

### A.2  Settings for GSM8k Experiments

Following Ye et al. (2025) and Nie et al. (2025), we use the widely-adopted lm-evaluation-harness[3] framework to evaluate the performance. For the GSM8K benchmark, we follow prior work and set the number of few-shot examples to 8 and the maximum length of answer to 256. We assess the GSM8K Accuracy and model efficiency of different models, where efficiency is defined by the inverse of the execution time measured on 8 Nvidia RTX 4090 GPUs with Huggingface's transformers library[4].

---

[3]https://github.com/EleutherAI/lm-evaluation-harness
[4]https://github.com/huggingface/transformers

In the figure, we report the relative accuracy and efficiency of MDMs compared to the baselines. These relative values are computed by dividing the actual accuracy and efficiency by those of the respective baselines – Qwen and the MDMs using the left-to-right strategy.

We list the detailed testing configurations for GSM8K in Appendix A.2.

| Models | Dream w/ entropy | Dream w/ left-to-right | LLaDA w/ low-confidence | LLaDA w/ left-to-right | Qwen2.5 |
|---|---|---|---|---|---|
| Max new tokens | 256 | 256 | 256 | 256 | 256 |
| Block length | N/A | N/A | 256 | 1 | N/A |
| Sampling algorithm | Entropy | Sequential | Low-confidence | Sequential | N/A |
| Sampling steps | {64,128,256} | N/A | {64,128,256} | 256 | N/A |
| Temperature | 0 | 0 | 0 | 0 | 0 |

Table 3: Testing Configurations for GSM8K

## A.3 Settings and Results for MBPP Experiments

For the MBPP experiment, the evaluation settings are similar to the GSM8K experiment. Specifically, use 3 few-shot examples and set the maximum answer length to 512, as suggested, to evaluate the MBPP Accuracy and efficiency. The definition of efficiency and the method for computing relative accuracy and efficiency values – by normalizing against the respective baselines – are identical to those used in the GSM8K evaluation.

The results are presented in Figure 2, and the detailed testing configurations for MBPP are listed in Appendix A.3

| Models | Dream w/ entropy | Dream w/ left-to-right | LLaDA w/ low-confidence | LLaDA w/ left-to-right | Qwen2.5 |
|---|---|---|---|---|---|
| Max new tokens | 512 | 512 | 512 | 512 | 512 |
| Block length | N/A | N/A | 512 | 1 | N/A |
| Sampling algorithm | Entropy | Sequential | Low-confidence | Sequential | N/A |
| Sampling steps | {128,256,512} | N/A | {128,256,512} | 512 | N/A |
| Temperature | 0.2 | 0.2 | 0 | 0 | 0 |
| Top p | 0.95 | 0.95 | N/A | N/A | N/A |

Table 4: Testing Configurations for MBPP

## A.4 Settings and Results for HumanEval Experiments

For the HumanEval experiment, we use zero-shot and set the maximum answer length to 512 to evaluate the HumanEval Accuracy and efficiency. The evaluation methods and definitions are identical to previous experiments.

The results are presented in Figure 3, and the detailed testing configurations for MBPP are listed in the table below.

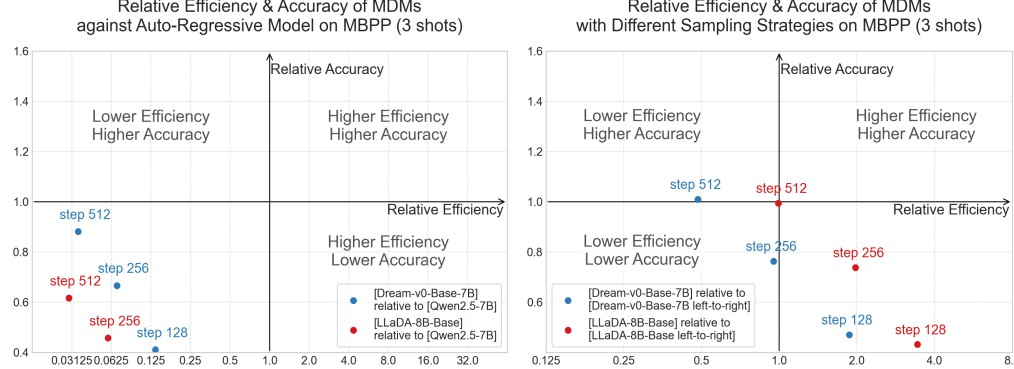

Figure 2: Efficiency and accuracy of MDMs on MBPP (3-shot): Similar to the GSM8K experiment, the left figure uses Qwen2.5-7B as the evaluation baseline and compares the MBPP relative accuracy and efficiency of Dream-v0-Base-7B and LLaDA-8B-Base across different sampling steps. The origin can be considered as the baseline result. The first quadrant demonstrates models that can simultaneously achieve higher efficiency and accuracy. From the figure, we can see that the MDMs all fall into the third quadrant, indicating lower efficiency and lower performance. The right figure uses a baseline in which the MDMs are constrained to generate the left-most masked token at each step, effectively mimicking auto-regressive decoding. Still, even under this setting, none of the configurations fall into the first quadrant.

| Models | Dream w/ entropy | Dream w/ left-to-right | LLaDA w/ low-confidence | LLaDA w/ left-to-right | Qwen2.5 |
|---|---|---|---|---|---|
| Max new tokens | 512 | 512 | 512 | 512 | 512 |
| Block length | N/A | N/A | 512 | 1 | N/A |
| Sampling algorithm | Entropy | Sequential | Low-confidence | Sequential | N/A |
| Sampling steps | {256,512} | N/A | {128,256,512} | 512 | N/A |
| Temperature | 0.2 | 0.2 | 0 | 0 | 0 |
| Top p | 0.95 | 0.95 | N/A | N/A | N/A |

Table 5: Testing Configurations for HumanEval

# B  Further Discussion

## B.1  Various Metrics in NLP Tasks.

Evaluation metrics in NLP tasks are inherently tied to the specific objectives and requirements of their respective domains. For general language modeling tasks, perplexity (Jelinek et al., 1977; Devlin et al., 2019) remains the metric of choice due to its ability to capture a model's predictive performance effectively. However, domain-specific tasks often demand more specialized evaluation criteria. For instance, in machine translation (Bahdanau, 2014; Wu et al., 2016), the BLEU score is widely regarded as a standard measure of translation quality (Papineni et al., 2002), while text generation tasks (Sutskever, 2014) frequently rely on metrics such as ROUGE to assess output fidelity (Lin, 2004). Similarly, tasks requiring reasoning (Wei et al., 2022c), such as mathematics (Bubeck et al., 2023) or code generation (Roziere et al., 2023; Ouyang et al., 2023), commonly adopt accuracy as an intuitive and straightforward measure of success.

## B.2  Discussion Regarding the Empirical Observations.

The above results reveal that MDMs can efficiently generate low-TER sentences but may incur higher costs when evaluating the generation under SER. One might think these results are contradictory. Note that several previous works have already shown that TER (a.k.a perplexity) may not reflect a model's true performance in solving several long-sequence understanding tasks (Huang

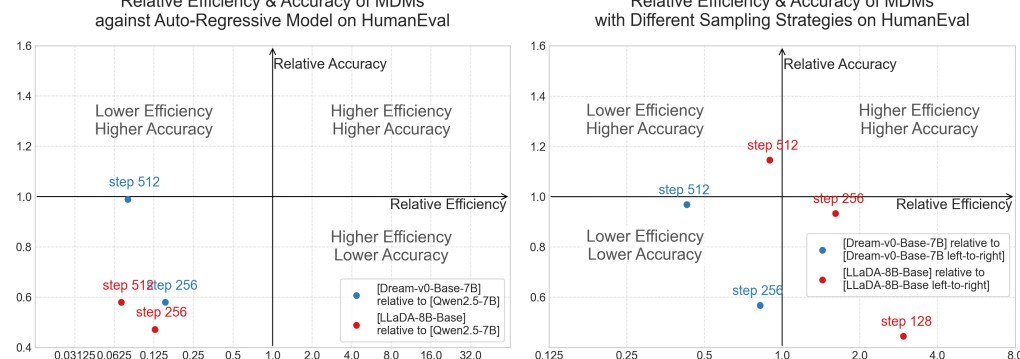

Figure 3: Efficiency and accuracy of MDMs on HumanEval: Similar to above, the left figure uses Qwen2.5-7B as the evaluation baseline and compares the HumanEval relative accuracy and efficiency of Dream-v0-Base-7B and LLaDA-8B-Base across different sampling steps. The origin can be considered as the baseline result. The first quadrant demonstrates models that can simultaneously achieve higher efficiency and accuracy. From the figure, we can see that the MDMs all fall into the third quadrant, indicating lower efficiency and lower performance. The right figure uses a baseline in which the MDMs are constrained to generate the left-most masked token at each step, effectively mimicking auto-regressive decoding. Still, even under this setting, none of the configurations fall into the first quadrant.

et al., 2022; Hu et al., 2024; Luden et al., 2024). Thus, it is natural to arrive at different conclusions depending on the metric used. Moreover, many practical scenarios have shown that the choice of evaluation metric significantly influences the conclusion of other problems (see Appendix B.1 for various metrics in NLP tasks). For instance, while the community has previously focused on the emergence phenomenon, recent works by Wei et al. (2022a) and Schaeffer et al. (2024) demonstrate that this phenomenon may stem from the use of non-smooth evaluation metrics.

## B.3 Extend to Efficient Sampling Strategies

In Sahoo et al. (2024) and Ou et al. (2024), an efficient sampling strategy `ddpm_cache` is proposed, which can reduce the sampling time by a constant order of magnitude. Specifically, this sampler is approximately 3-4 times faster than previously used samplers when the number of sampling steps is large. In this section, we discuss the influence of `ddpm_cache` on our conclusions under different sampling steps.

First, we briefly introduce the principles of `ddpm_cache`. It utilizes the observation that if no locations are sampled at a given step, the sequence remains unchanged. Consequently, when the reverse model is not conditioned on time, the cached value computed during the first time this sequence went through the reverse model can be reused, instead of going through the reverse model again.

This sampling strategy does not affect our main theorems, as they are based solely on the sampled locations at each step, while unsampled locations are not considered. As for the evaluation metrics for computational efficiency in our experiments, we break it down into the following two cases:

1. When the number of sampling steps is much smaller than the sequence length, which is the primary scenario we focus on, the expectation of steps where no new locations are sampled is relatively low, resulting in a computational cost that is nearly linear with respect to the number of sampling steps.

2. As the number of sampling steps becomes larger, the computational cost is mainly dependent on the number of valid steps where at least one location is sampled. As a matter of fact, the expectation of the number of valid steps increases as the number of sampling steps increases, and the maximum number of valid steps is equal to the number of sampling steps. In this case, the MDMs offer no computational advantage over auto-regressive models.

Based on the above conclusions, we can find that for tasks requiring a low TER, using `ddpm_cache` can further accelerate the generation of MDMs, suggesting high efficiency. Conversely, for tasks that

demand a low SER, we have shown that the number of sampling steps need to be large enough, such that MDMs can not generate with low cost even when using `ddpm_cache`. Therefore, we extend our findings to MDMs with efficient sampling strategies.

## C  Auxiliary Lemma

In this section, we present some technical lemmas for the proof of our main results.

**Lemma C.1** (Upper Bound for Multi-tokens Sampling). *Let $\boldsymbol{X} = (X_1, X_2, \ldots, X_k) \in [N]^k$ be a random vector following the distribution $q$, where each component $X_i$ follows the marginal distribution $q_i$. Define $\tilde{\boldsymbol{X}} = (\tilde{X}_1, \tilde{X}_2, \ldots, \tilde{X}_k) \sim p$ as another random vector, where the components $\tilde{X}_i$ are sampled independently according to $p_i$. Let $\delta = \max_i \{D_{\mathrm{KL}}(q_i \| p_i)\}$, then, the KL divergence between $p$ and $q$ satisfies the inequality:*

$$D_{\mathrm{KL}}(q\|p) \leq (k-1)\log N + k\delta.$$

*Proof.* Using the chain rule for probabilities, the KL divergence can be written as:

$$D_{\mathrm{KL}}(q\|p) = \mathbb{E}_q\left[\sum_{i=1}^{k} \log\left(\frac{q_i(x_i \mid \mathbf{x}_{<i})}{p_i(x_i)}\right)\right],$$

where $\mathbf{x}_{<i} = (x_1, \ldots, x_{i-1})$. For $i = 1$, there are no preceding variables, so:

$$\mathbb{E}_q\left[\log\left(\frac{q_1(x_1)}{p_1(x_1)}\right)\right] = D_{\mathrm{KL}}(q_1\|p_1).$$

For $i > 1$, we bound:

$$\mathbb{E}_q\left[\log\left(\frac{q_i(x_i \mid \mathbf{x}_{<i})}{p_i(x_i)}\right)\right] \leq \mathbb{E}_q\left[\log\left(\frac{1}{p_i(x_i)}\right)\right].$$

Decomposing $\mathbb{E}_q\left[\log\left(1/p_i(x_i)\right)\right]$, we get:

$$\mathbb{E}_q\left[\log\left(\frac{1}{p_i(x_i)}\right)\right] = \mathbb{E}_q\left[\log\left(\frac{q_i(x_i)}{p_i(x_i)}\right)\right] + \mathbb{E}_q\left[\log\left(\frac{1}{q_i(x_i)}\right)\right].$$

The first term is $D_{\mathrm{KL}}(q_i\|p_i)$, and the second term is $-\mathbb{E}_q\left[\log q_i(x_i)\right]$, which represents the entropy of $q_i$. Since the entropy of any distribution over $[N]$ is at most $\log N$, we have:

$$-\mathbb{E}_q\left[\log q_i(x_i)\right] \leq \log N.$$

Thus:

$$\mathbb{E}_q\left[\log\left(\frac{q_i(x_i \mid \mathbf{x}_{<i})}{p_i(x_i)}\right)\right] \leq D_{\mathrm{KL}}(q_i\|p_i) + \log N.$$

Summing over all $i = 1, \ldots, k$, we obtain:

$$D_{\mathrm{KL}}(q\|p) = \sum_{i=1}^{k} \mathbb{E}_q\left[\log\left(\frac{q_i(x_i \mid \mathbf{x}_{<i})}{p_i(x_i)}\right)\right] \leq D_{\mathrm{KL}}(q_1\|p_1) + \sum_{i=2}^{k} \left(D_{\mathrm{KL}}(q_i\|p_i) + \log N\right).$$

Reorganizing, we have:

$$D_{\mathrm{KL}}(q\|p) \leq \sum_{i=1}^{k} D_{\mathrm{KL}}(q_i\|p_i) + (k-1)\log N.$$

Since $D_{\mathrm{KL}}(q_i\|p_i) \leq \delta$ for all $i$, the total sum of marginal KL divergences is bounded by $k\delta$. Therefore:

$$D_{\mathrm{KL}}(q\|p) \leq k\delta + (k-1)\log N.$$

This completes the proof.

$\square$

**Lemma C.2** (Chernoff Bound). *Let $X_1, \ldots, X_n$ be independent random variables taking values in $\{0, 1\}$. Define $X = \sum_{i=1}^{n} X_i$ as the sum of these independent random variables, and let $\mu = \mathbb{E}[X]$ denote the expected value of $X$. Then, the following probabilistic bounds hold:*

$$\Pr(X \geq (1 + \delta)\mu) \leq e^{-\frac{\delta^2 \mu}{2+\delta}}, \qquad \text{for } \delta \geq 0,$$

$$\Pr(X \leq (1 - \delta)\mu) \leq e^{-\frac{\delta^2 \mu}{2}}, \qquad \text{for } 0 < \delta < 1.$$

**Lemma C.3** (Pinsker's Inequality). *Let $p$ and $q$ be two probability distributions. Then, the total variation distance between these distributions satisfies:*

$$D_{\mathrm{TV}}(p, q) \leq \sqrt{\frac{1}{2} D_{\mathrm{KL}}(p \| q)}.$$

*Specifically, since $D_{\mathrm{TV}}(p, q) = \frac{1}{2} \|p - q\|_1$, the following inequality holds:*

$$\|p - q\|_1 \leq \sqrt{2 D_{\mathrm{KL}}(p \| q)}.$$

# D    Proof for Theorem 4.2

This section provides the complete proof of Theorem 4.2. We first outline the proof strategy, then present the detailed arguments with supporting lemmas and definitions.

Our proof rests on reformulating TER bounds through KL divergence and carefully analyzing dependencies in the n-gram setting. The key steps are:

- We establish a connection between the perplexity of the discrete diffusion model and the KL divergence between generated and data distributions. This involves deriving an upper bound on KL divergence using expected divergence over reverse processes (Lemma D.2) and decomposing this divergence into per-step conditional KL terms (Lemma D.3).

- We analyze n-gram model dependencies through a rigorous characterization of reverse processes (Definition D.1) and separators $(n - 1)$ continuous sampled tokens that create independent intervals (Definition D.4). This leads to a precise formulation of per-step dependencies using these separators (Definition D.5).

- We derive an upper bound for the KL divergence between generated and data distributions based on the number of per-step dependencies (Lemmas D.8 and D.9).

- We employ probabilistic bounds to analyze and bound the number of per-step dependencies (Lemmas C.1, D.10 and D.11).

- Finally, we demonstrate the existence of a schedule achieving small KL divergence with $O(\frac{n-1}{\epsilon^n})$ steps by constructing an efficient sampling schedule using the preceding lemmas (Lemma D.12).

To begin the formal proof, we introduce key definitions for analyzing the discrete diffusion process. Consider a masking schedule $\alpha_t$ and a sequence of sampling time steps $t_i = \frac{N-i}{N}$. For a sequence of length $L$, we define an instance of the discretization of the reverse process $\tau$ as follows:

**Definition D.1** (An Instance of Reverse Process). Let $\tau = (\mathcal{M}_1, \mathcal{M}_2, \ldots, \mathcal{M}_N)$ represent a reverse process, where $\mathcal{M}_i = \{l_{ij}\}$ denotes the set of locations sampled at step $i$. For a sequence of length $L$, the sets $\mathcal{M}_i$ satisfy the following conditions:

$$\bigcup_{i \in [N]} \mathcal{M}_i = [L] \quad \text{and} \quad \bigcap_{i \in [N]} \mathcal{M}_i = \emptyset.$$

Specifically, we denote $\mathcal{M}_{<i}$ as the union of all locations sampled prior to step $t_i$:

$$\mathcal{M}_{<i} = \bigcup_{j < i} \mathcal{M}_j.$$

Under a given instance of the reverse process $\tau$, at each time step $t_i = \frac{N-i}{N}$, the set of locations $\mathcal{M}_i = \{l_{ij}\}$ is sampled. Let $\tilde{x}_i$ denote the tokens associated with the locations sampled at time step $t_i$. Given the masking schedule $\alpha_t$, there exist multiple possible instances of the reverse process. We denote the distribution over these instances by $\mathrm{REVR}(\alpha_t, N, L)$.

**Lemma D.2** (KL Divergence Upper Bound for the Masking Schedule). *Let $q$ denote the data distribution over sequences of length $L$, and let $p$ denote the distribution over sequences of length $L$ generated by the reverse model $p_\theta$ with masking schedule $\alpha_t$ and $N$ sampling steps. The KL divergence between $q$ and $p$ satisfies the following upper bound:*

$$D_{\mathrm{KL}}(q\|p) \leq \mathbb{E}_{\tau \sim \mathrm{REVR}(\alpha_t, N, L)} D_{\mathrm{KL}}(q\|p(\cdot|\tau)),$$

*where the expectation is taken over the distribution of reverse processes $\tau$ induced by $\mathrm{REVR}(\alpha_t, N, L)$.*

*Proof.* Let $\mathcal{X}$ denote the set of all possible generated sequences. Then, the KL divergence between $q$ and $p$ is given by:

$$D_{\mathrm{KL}}(q\|p) = \sum_{\boldsymbol{x} \in \mathcal{X}} q(\boldsymbol{x}) \log \frac{q(\boldsymbol{x})}{p(\boldsymbol{x})}.$$

Let $h$ denote the distribution over reverse processes $\tau \sim \mathrm{REVR}(\alpha_t, N, L)$. Due to the convexity of $\log \frac{1}{x}$, by applying Jensen's inequality, we can obtain:

$$\log \frac{1}{p(\boldsymbol{x})} = \log \frac{1}{\sum_\tau h(\tau) \cdot p(\boldsymbol{x}|\tau)} \leq \sum_\tau h(\tau) \cdot \log \frac{1}{p(\boldsymbol{x}|\tau)}.$$

Since data distribution $q$ is independent of reverse process $\tau$:

$$q(\boldsymbol{x}) = q(\boldsymbol{x}|\tau), \quad \forall \tau.$$

Therefore, we have:

$$\log \frac{q(\boldsymbol{x})}{p(\boldsymbol{x})} \leq \sum_\tau h(\tau) \log \frac{q(\boldsymbol{x}|\tau)}{p(\boldsymbol{x}|\tau)}.$$

Substituting this back, we can get the final result:

$$
\begin{aligned}
D_{\mathrm{KL}}(q\|p) &\leq \sum_{\boldsymbol{x} \in \mathcal{X}} \sum_\tau h(\tau) q(\boldsymbol{x}) \log \frac{q(\boldsymbol{x}|\tau)}{p(\boldsymbol{x}|\tau)} \\
&= \sum_\tau h(\tau) \sum_{\boldsymbol{x} \in \mathcal{X}} q(\boldsymbol{x}|\tau) \log \frac{q(\boldsymbol{x}|\tau)}{p(\boldsymbol{x}|\tau)} \\
&= \mathbb{E}_{\tau \sim \mathrm{REVR}(\alpha_t, N, L)} D_{\mathrm{KL}}(q(\cdot|\tau)\|p(\cdot|\tau)) \\
&= \mathbb{E}_{\tau \sim \mathrm{REVR}(\alpha_t, N, L)} D_{\mathrm{KL}}(q\|p(\cdot|\tau)).
\end{aligned}
$$

$\square$

We next establish an upper bound for the KL divergence between the distribution of sequences sampled under an instance of the reverse process $\tau$ and the ground-truth distribution in the $n$-gram setting. To achieve this, we leverage the chain rule for KL divergence, which allows decomposition of the KL divergence of the entire sequence into a summation of the KL divergences at each individual step of the process.

**Lemma D.3** (KL Divergence Decomposition for the Reverse Process). *Consider an instance of reverse process $\tau = (\mathcal{M}_1, \mathcal{M}_2, \ldots, \mathcal{M}_N) \sim \mathrm{REVR}(\alpha_t, N, L)$. Let $\tilde{\boldsymbol{x}}_i$ denote the set of tokens corresponding to the locations sampled at time step $t_i$, and $\tilde{\boldsymbol{x}}_{<i}$ denote the set of tokens sampled at all steps prior to step $t_i$. The KL divergence between the ground-truth distribution $q$ and the distribution $p_\tau$ generated by the reverse process $\tau$ and reverse model $p_\theta$ satisfies the following decomposition:*

$$D_{\mathrm{KL}}(q\|p_\tau) = \sum_{i=1}^N \mathbb{E}_{\tilde{\boldsymbol{x}}_{<i}} D_{\mathrm{KL}}(q(\tilde{\boldsymbol{x}}_i|\tilde{\boldsymbol{x}}_{<i})\|p_\tau(\tilde{\boldsymbol{x}}_i|\tilde{\boldsymbol{x}}_{<i})),$$

*Proof.* Given the reverse process $\tau$, the reverse model samples $\tilde{\boldsymbol{x}}_i$ sequentially from $i = 1$ to $N$, and the probability of sampling $\tilde{\boldsymbol{x}}_i$ at step $t_i$ depends only on the previously sampled tokens $\tilde{\boldsymbol{x}}_{<i}$. Therefore, the distribution $p(\boldsymbol{x})$ can be factorized as:

$$p_\tau(\boldsymbol{x}) = \prod_{i=1}^N p_\tau(\tilde{\boldsymbol{x}}_i|\tilde{\boldsymbol{x}}_{<i}).$$

On the other hand, since the data distribution $q$ is independent of the reverse process $\tau$, it can similarly be decomposed as:

$$q(\boldsymbol{x}) = \prod_{i=1}^{N} q(\tilde{\boldsymbol{x}}_i | \tilde{\boldsymbol{x}}_{<i}).$$

Applying the chain rule for KL divergence, we obtain:

$$D_{\mathrm{KL}}(q\|p_\tau) = \sum_{i=1}^{N} \mathbb{E}_{\tilde{\boldsymbol{x}}_{<i}} D_{\mathrm{KL}}(q(\tilde{\boldsymbol{x}}_i | \tilde{\boldsymbol{x}}_{<i}) \| p_\tau(\tilde{\boldsymbol{x}}_i | \tilde{\boldsymbol{x}}_{<i})).$$

$\square$

Next, we derive an upper bound for the KL divergence at each step of the reverse process. In the $n$-gram setting, it is important to note that, at step $t_i$, the tokens $x_{l_{ij}}$ and $x_{l_{ij'}}$ are conditionally independent if there are at least $n-1$ consecutive tokens between the positions $l_{ij}$ and $l_{ij'}$ that have already been sampled prior to step $i$. Under this condition, sampling these two tokens simultaneously incurs no sampling error, as the distributions of $x_{l_{ij}}$ and $x_{l_{ij'}}$ are independent.

To formalize this concept, we introduce a measure of dependencies among the tokens sampled in $\mathcal{M}_i$ during the reverse process. For the $i$-th reverse step in the $n$-gram setting, the number of dependencies, denoted as $\mathrm{DEP}_n(\mathcal{M}_i, \mathcal{M}_{<i})$, is determined by the structure of $\mathcal{M}_{<i}$. Specifically, it depends on the number of separators in $\mathcal{M}_{<i}$, denoted as $\mathrm{SEP}_n(\mathcal{M}_{<i})$, as described in the following definition.

**Definition D.4** (Number of Separators in a Reverse Step). Consider a reverse process $\tau = (\mathcal{M}_1, \mathcal{M}_2, \ldots, \mathcal{M}_N)$, where $\mathcal{M}_{<i} = \bigcup_{j<i} \mathcal{M}_j$ represents the union of all previously sampled location sets. The set $\mathcal{M}_{<i}$ can be partitioned into several contiguous segments. Let $\mathcal{S}_1, \mathcal{S}_2, \cdots, \mathcal{S}_k$ denote the segments containing at least $n-1$ consecutive tokens (i.e., $|\mathcal{S}_j| \geq n-1$) with the maximum $k$. We refer to these segments as separators, and denote the number of separators in the set $\mathcal{M}_{<i}$ as:

$$\begin{aligned} \mathrm{SEP}_n(\mathcal{M}_{<i}) = \max \quad & k \\ \text{s.t.} \quad & |\mathcal{S}_j| \geq n-1, \ \mathcal{S}_j \subset \mathcal{M}_{<i}, \quad \forall j \in [k], \\ & \mathcal{S}_j \cap \mathcal{S}'_j = \emptyset, \quad \forall j \neq j'. \end{aligned}$$

Note that if a contiguous segment $\mathcal{S}$ in $\mathcal{M}_{<i}$ contains at least $d(n-1)$ consecutive tokens, where $d$ is an integer, then $\mathcal{S}$ consists of at least $d$ separators.

**Definition D.5** (Number of Dependencies in a Reverse Step). Consider a reverse process $\tau = (\mathcal{M}_1, \mathcal{M}_2, \ldots, \mathcal{M}_N)$. The separators of $\mathcal{M}_{<i}$ divide the sequence into at most $\mathrm{SEP}_n(\mathcal{M}_{<i}) + 1$ disjoint intervals $\mathcal{I}_1, \mathcal{I}_2, \ldots, \mathcal{I}_k$. Under the $n$-gram setting, the sampling within each interval is independent of the sampling in other intervals. The number of dependencies of step $t_i$ is defined as the number of intervals $\mathcal{I}_p$ (for $p = 1, \ldots, k$) that contain at least one location in $\mathcal{M}_i$:

$$\mathrm{DEP}_n(\mathcal{M}_i, \mathcal{M}_{<i}) = |\mathcal{M}_i| - \sum_{p=1}^{k} \mathbb{I}\left[\mathcal{I}_p \cap \mathcal{M}_i \neq \emptyset\right],$$

where $\mathbb{I}$ is the indicator function.

To illustrate this definition, we provide the following example:

**Example D.6** (Computing Dependencies in the $n$-gram Setting). Consider a token sequence of length 10, denoted as $\boldsymbol{x} = (x_1, x_2, \ldots, x_{10})$, with $n = 4$. Let the previously sampled location set be $\mathcal{M}_{<i} = \{2, 3, 4, 6, 7\}$ and the current location set be $\mathcal{M}_i = \{1, 5, 9\}$.

1. **Identify contiguous segments in $\mathcal{M}_{<i}$ containing at least $n-1 = 3$ consecutive tokens:** The set $\mathcal{M}_{<i} = \{2, 3, 4, 6, 7\}$ forms the following contiguous segments:

$$\{2, 3, 4\} \quad \text{and} \quad \{6, 7\}.$$

Only the segment $\{2, 3, 4\}$ contains at least $n-1 = 3$ consecutive tokens. Thus, we have $\mathcal{S}_1 = \{2, 3, 4\}$. The sequence is then divided into the following disjoint intervals:

$$\mathcal{I}_1 = \{1\}, \quad \mathcal{I}_2 = \{5, 6, 7, 8, 9, 10\}.$$

2. **Determine which intervals overlap with $\mathcal{M}_i = \{1, 5, 9\}$:** Token 1 belongs to interval $\mathcal{I}_1$, and tokens 5 and 9 belong to interval $\mathcal{I}_2$.

3. **Compute the number of dependencies:** The number of dependencies is:

$$\mathrm{DEP}_n(\mathcal{M}_i, \mathcal{M}_{<i}) = |\mathcal{M}_i| - \sum_{p=1}^{k} \mathbb{I}[\mathcal{I}_p \cap \mathcal{M}_i \neq \emptyset] = 3 - 2 = 1.$$

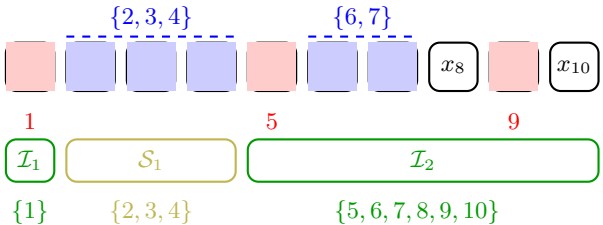

Figure 4: Illustration of the example for computing dependencies in the $n$-gram setting. Tokens $x_2, x_3, x_4, x_6, x_7$ (blue) represent the previously sampled location set $\mathcal{M}_{<i}$, forming two contiguous segments: $\{2, 3, 4\}$ and $\{6, 7\}$. The current sampled locations $x_1, x_5, x_9$ (red) overlap with disjoint intervals $\mathcal{I}_1 = \{1\}$ and $\mathcal{I}_2 = \{5, 6, 7, 8, 9, 10\}$. The number of dependencies is computed as $\mathrm{DEP}_n(\mathcal{M}_i, \mathcal{M}_{<i}) = |\mathcal{M}_i| - (\text{number of overlapping intervals}) = 3 - 2 = 1$.

This example demonstrates how dependencies are computed, highlighting the interaction between previously sampled locations and the current reverse step. Such formalization is critical for understanding the efficiency and accuracy of discrete diffusion processes.

Finally, we extend this concept to define the total number of dependencies across an entire reverse process:

**Definition D.7** (Number of Dependencies in a Reverse Process). Consider a reverse process $\tau = (\mathcal{M}_1, \mathcal{M}_2, \ldots, \mathcal{M}_N)$. Under the $n$-gram setting, the total number of dependencies in the process is defined as the sum of the dependencies across all steps:

$$\mathrm{DEP}_n(\tau) = \sum_{i=1}^{N} \mathrm{DEP}_n(\mathcal{M}_i, \mathcal{M}_{<i}).$$

Using the definition of $\mathrm{DEP}_n(\tau)$, we can bound the KL divergence between the distribution of sequences sampled under an instance of the reverse process and the ground-truth distribution in the $n$-gram setting.

**Lemma D.8** (KL Divergence Upper Bound for the Instance of Reverse Process). *Let $q$ denote the data distribution for sequences of length $L$, and let $p$ denote the distribution of sequences of length $L$ generated by reverse model $p_\theta$ via the reverse process $\tau$. Under Assumption 4.1, the following upper bound holds:*

$$D_{\mathrm{KL}}(q \| p(\cdot | \tau)) \leq \mathrm{DEP}_n(\tau) \log |\mathcal{V}| + L\epsilon_{\mathrm{learning}},$$

*where $\mathcal{V}$ denote the vocabulary.*

*Proof.* Using Lemma D.3, we have:

$$D_{\mathrm{KL}}(q \| p_\tau) = \sum_{i=1}^{N} \mathbb{E}_{\tilde{\boldsymbol{x}}_{<i}} D_{\mathrm{KL}}(q(\tilde{\boldsymbol{x}}_i | \tilde{\boldsymbol{x}}_{<i}) \| p_\tau(\tilde{\boldsymbol{x}}_i | \tilde{\boldsymbol{x}}_{<i})).$$

For each time step $t_i$:

$$\mathbb{E}_{\tilde{\boldsymbol{x}}_{<i}} D_{\mathrm{KL}}(q(\tilde{\boldsymbol{x}}_i | \tilde{\boldsymbol{x}}_{<i}) \| p_\tau(\tilde{\boldsymbol{x}}_i | \tilde{\boldsymbol{x}}_{<i})) = \mathbb{E}_{\tilde{\boldsymbol{x}}_{<i}} \sum_{\tilde{\boldsymbol{x}}_i \in \mathcal{V}^{|\mathcal{M}_i|}} q(\tilde{\boldsymbol{x}}_i | \tilde{\boldsymbol{x}}_{<i}) \log \frac{q(\tilde{\boldsymbol{x}}_i | \tilde{\boldsymbol{x}}_{<i})}{p_\tau(\tilde{\boldsymbol{x}}_i | \tilde{\boldsymbol{x}}_{<i})}.$$

Given $\mathcal{M}_i$ and $\mathcal{M}_{<i}$, the tokens $\tilde{\boldsymbol{x}}_i$ at step $t_i$ can be partitioned into independently sampled token sets $\tilde{\boldsymbol{x}}_i^{(1)}, \cdots, \tilde{\boldsymbol{x}}_i^{(m)}$ with $k_j$ denoting the size of each token set:

$$k_j = |\tilde{\boldsymbol{x}}_i^{(j)}|, \ j \in [m], \quad m = |\mathcal{M}_i| - \mathrm{DEP}_n(\mathcal{M}_i, \mathcal{M}_{<i}).$$

Using the independence, for each $\tilde{\boldsymbol{x}}_{<i}$, we can decompose the sum into:

$$\sum_{\tilde{\boldsymbol{x}}_i \in \mathcal{V}^{|\mathcal{M}_i|}} q(\tilde{\boldsymbol{x}}_i|\tilde{\boldsymbol{x}}_{<i}) \log \frac{q(\tilde{\boldsymbol{x}}_i|\tilde{\boldsymbol{x}}_{<i})}{p_\tau(\tilde{\boldsymbol{x}}_i|\tilde{\boldsymbol{x}}_{<i})} = \sum_{j=1}^{m} \sum_{\tilde{\boldsymbol{x}}_i^{(j)} \in \mathcal{V}^{k_j}} q(\tilde{\boldsymbol{x}}_i^{(j)}|\tilde{\boldsymbol{x}}_{<i}) \log \frac{q(\tilde{\boldsymbol{x}}_i^{(j)}|\tilde{\boldsymbol{x}}_{<i})}{p_\tau(\tilde{\boldsymbol{x}}_i^{(j)}|\tilde{\boldsymbol{x}}_{<i})}$$

$$= \sum_{j=1}^{m} D_{\mathrm{KL}}(q(\tilde{\boldsymbol{x}}_i^{(j)}|\tilde{\boldsymbol{x}}_{<i}) \| p_\tau(\tilde{\boldsymbol{x}}_i^{(j)}|\tilde{\boldsymbol{x}}_{<i})).$$

Under Assumption 4.1, the KL divergence between $q$ and $p_\theta$ is bounded by:

$$D_{\mathrm{KL}}(q_{0|t}(x_0^i \mid \boldsymbol{x}_t) \| p_\theta(x_0^i \mid \boldsymbol{x}_t)) < \epsilon_{\text{learning}}, \quad \forall \, t \text{ and } \boldsymbol{x}_t.$$

By Lemma C.1, we know that:

$$D_{\mathrm{KL}}(q(\tilde{\boldsymbol{x}}_i^{(j)}|\tilde{\boldsymbol{x}}_{<i}) \| p_\tau(\tilde{\boldsymbol{x}}_i^{(j)}|\tilde{\boldsymbol{x}}_{<i}) \le (k_j - 1) \log |\mathcal{V}| + k_j \epsilon_{\text{learning}}.$$

Substituting back:

$$\sum_{\tilde{\boldsymbol{x}}_i \in \mathcal{V}^{|\mathcal{M}_i|}} q(\tilde{\boldsymbol{x}}_i|\tilde{\boldsymbol{x}}_{<i}) \log \frac{q(\tilde{\boldsymbol{x}}_i|\tilde{\boldsymbol{x}}_{<i})}{p_\tau(\tilde{\boldsymbol{x}}_i|\tilde{\boldsymbol{x}}_{<i})} \le \sum_{j=1}^{m} (k_j - 1) \log |\mathcal{V}| + k_j \epsilon_{\text{learning}}.$$

Using the fact that

$$\sum_{j=1}^{m} (k_j - 1) = |\mathcal{M}_i| - m = \mathrm{DEP}_n(\mathcal{M}_i, \mathcal{M}_{<i}), \quad \sum_{j=1}^{m} k_j = |\mathcal{M}_i|$$

we can obtain:

$$\mathbb{E}_{\tilde{\boldsymbol{x}}_{<i}} D_{\mathrm{KL}}(q(\tilde{\boldsymbol{x}}_i|\tilde{\boldsymbol{x}}_{<i}) \| p_\tau(\tilde{\boldsymbol{x}}_i|\tilde{\boldsymbol{x}}_{<i})) \le \mathrm{DEP}_n(\mathcal{M}_i, \mathcal{M}_{<i}) \log |\mathcal{V}| + |\mathcal{M}_i| \epsilon_{\text{learning}}.$$

Thus, combined with the definition of $\mathrm{DEP}_n(\tau)$ and $p_\tau = p(\cdot|\tau)$, we can draw the final conclusion:

$$D_{\mathrm{KL}}(q\|p(\cdot|\tau)) \le \sum_{i=1}^{N} (\mathrm{DEP}_n(\mathcal{M}_i, \mathcal{M}_{<i}) \log |\mathcal{V}| + |\mathcal{M}_i| \epsilon_{\text{learning}})$$

$$= \mathrm{DEP}_n(\tau) \log |\mathcal{V}| + L \epsilon_{\text{learning}}.$$

$\square$

The above Lemma directly leads to the bound for the KL divergence between the distribution of sequences generated by the reverse model with a given masking schedule and the ground-truth distribution in the $n$-gram setting.

**Lemma D.9** (KL Divergence Upper Bound for a Masking Schedule). *Let $q$ denote the data distribution over sequences of length $L$, and let $p$ denote the distribution over sequences of length $L$ generated by the reverse model $p_\theta$ with masking schedule $\alpha_t$ and $N$ reverse steps. Under Assumption 4.1, the KL divergence between $q$ and $p$ satisfies the following upper bound:*

$$D_{\mathrm{KL}}(q\|p) \le \log |\mathcal{V}| \sum_{i=1}^{N} \mathbb{E}_{\tau \sim \mathrm{REVR}(L, \alpha_t, N)} \mathrm{DEP}_n(\mathcal{M}_i, \mathcal{M}_{<i}) + L \epsilon_{\text{learning}}.$$

*Proof.* By Lemma D.2, we can obtain:

$$D_{\mathrm{KL}}(q\|p) \le \mathbb{E}_{\tau \sim \mathrm{REVR}(\alpha_t, N, L)} D_{\mathrm{KL}}(q\|p(\cdot|\tau)).$$

Applying Lemma D.8 to the instances of reverse process, we can conclude that:

$$D_{\mathrm{KL}}(q\|p) \le \mathbb{E}_{\tau \sim \mathrm{REVR}(\alpha_t, N, L)} \sum_{i=1}^{N} \mathrm{DEP}_n(\mathcal{M}_i, \mathcal{M}_{<i}) \log |\mathcal{V}| + L \epsilon_{\text{learning}}$$

$$= \log |\mathcal{V}| \sum_{i=1}^{N} \mathbb{E}_{\tau \sim \mathrm{REVR}(L, \alpha_t, N)} \mathrm{DEP}_n(\mathcal{M}_i, \mathcal{M}_{<i}) + L \epsilon_{\text{learning}}.$$

$\square$

For the final estimation, we need to derive an upper bound for the expected number of dependencies at each reverse step. First, we use Chernoff Bound to control the number of separators and new locations at each reverse step for a given masking schedule.

**Lemma D.10** (Bounds on Separator and New Location Count at Each Reverse Step). *Given a sequence of length $L$, a masking schedule $\alpha_t$, and $N$ reverse steps. Assume that $L$ is divisible by $n-1$. Given the time step $t_i = \frac{N-i}{N}$, let* NEW *denote the number of locations sampled at step $t_i$, and* $\mathrm{SEP}_n$ *denote the number of separators in the previously sampled locations. Under the $n$-gram setting, the following bounds hold for* NEW *and* $\mathrm{SEP}_n$:

$$\Pr\left(\mathrm{SEP}_n \leq \frac{Lp_i^{n-1}}{2(n-1)}\right) \leq e^{-\frac{Lp_i^{n-1}}{8(n-1)}},$$

$$\Pr\left(\mathrm{NEW} \geq 2L\delta_i\right) \leq e^{-\frac{L\delta_i}{3}},$$

*where $p_i = \alpha_{t_{i-1}}$ and $\delta_i = \alpha_{t_i} - \alpha_{t_{i-1}}$.*

*Proof.* Given a masking schedule $\alpha_t$, using the expression of true reverse process in Equation (2) and $\alpha_1 = 0$, we can compute the probability $p^{(i)}$ of a token being sampled at time step $t_i$ to be:

$$p^{(i)} = \frac{\alpha_{t_i} - \alpha_{t_{i-1}}}{1 - \alpha_{t_{i-1}}} \cdot \prod_{j=1}^{i-1} \frac{1 - \alpha_{t_j}}{1 - \alpha_{t_{j-1}}} = \alpha_{t_i} - \alpha_{t_{i-1}} = \delta_i.$$

Therefore, $\delta_i$ is the probability of a location being sampled at time step $t_i$. Summing up $\delta_i$, we can know that $p_i = \sum_{j=1}^{i-1} \delta_j$ is the probability of a location being sampled prior to time step $t_i$.

To derive a bound for $\mathrm{SEP}_n$, we partition the sequence into $\frac{L}{n-1}$ intervals, each of length $n-1$. For a given interval, the probability that all locations within the interval have been sampled prior to step $t_i$ is $p_i^{n-1}$. Define $X_j = 1$ if the locations in the $j$-th interval have been sampled prior to $t_i$, and $X_j = 0$ otherwise. The random variables $X_1, X_2, \cdots, X_{\frac{L}{n-1}}$ are independent and satisfy the following expectation:

$$\mathbb{E}_{\tau \sim \mathrm{REVR}(L, \alpha_t, N)} \sum_{j=1}^{\frac{L}{n-1}} X_j = \frac{Lp_i^{n-1}}{n-1}.$$

By the definition of $\mathrm{SEP}_n$, we know that:

$$\mathrm{SEP}_n \geq \sum_{j=1}^{\frac{L}{n-1}} X_j.$$

Applying Lemma C.2 to the sum of $X_j$, we derive:

$$\Pr\left(\mathrm{SEP}_n \leq \frac{Lp_i^{n-1}}{2(n-1)}\right) \leq \Pr\left(\sum_{j=1}^{\frac{L}{n-1}} X_j \leq \frac{Lp_i^{n-1}}{2(n-1)}\right) \leq e^{-\frac{Lp_i^{n-1}}{8(n-1)}}.$$

Next, we consider the bound for NEW. Given that the sequence contains $L$ locations and the probability of sampling any specific location at step $t_i$ is $\delta_i$, the expected number of new locations sampled at $t_i$ is given by:

$$\mathbb{E}_{\tau \sim \mathrm{REVR}(L, \alpha_t, N)} \mathrm{NEW} = L\delta_i.$$

Since the sampling of each location occurs independently, applying Lemma C.2, we have:

$$\Pr\left(\mathrm{NEW} \geq 2L\delta_i\right) \leq e^{-\frac{L\delta_i}{3}}.$$

$\square$

Using the above lemma, we can divide the estimation for the number of dependencies into three cases, and derive the bound case by case. This is achieved by using a variety of means and careful estimations.

**Lemma D.11** (Upper Bound for the Expectation of Dependencies at Each Reverse Step). *Given a sequence of length L, a masking schedule $\alpha_t$, and N reverse steps. Assume $L\delta_i > 1$, then the expected number of dependencies at time step $t_i = \frac{N-i}{N}$ satisfies:*

$$\mathbb{E}_{\tau \sim \text{REVR}(L, \alpha_t, N)} \text{DEP}_n(\mathcal{M}_i, \mathcal{M}_{<i}) \leq \frac{9}{3 + L\delta_i} + \frac{C(n-1)L\delta_i^2}{p_i^{n-1}},$$

*where $p_i = \alpha_{t_{i-1}}$, $\delta_i = \alpha_{t_i} - \alpha_{t_{i-1}}$, and C is a constant.*

*Proof.* By Lemma D.10, at step $t_i$, the following bounds hold:

$$\Pr\left(\text{SEP}_n \leq \frac{Lp_i^{n-1}}{2(n-1)}\right) \leq e^{-\frac{Lp_i^{n-1}}{8(n-1)}},$$

$$\Pr\left(\text{NEW} \geq 2L\delta_i\right) \leq e^{-\frac{L\delta_i}{3}}.$$

Since $\text{DEP}_n(\mathcal{M}_i, \mathcal{M}_{<i}) \geq 0$, its expectation can be decomposed into three components:

$$\mathbb{E}_{\tau \sim \text{REVR}(L, \alpha_t, N)} \text{DEP}_n(\mathcal{M}_i, \mathcal{M}_{<i}) = \Pr\left(\text{NEW} \geq 2L\delta_i\right) \cdot \mathbb{E}_{\substack{\tau \sim \text{REVR}(L, \alpha_t, N) \\ \text{NEW} \geq 2L\delta_i}} \text{DEP}_n(\mathcal{M}_i, \mathcal{M}_{<i}) \quad \textbf{(Case 1)}$$

$$+ \Pr\left(\text{SEP}_n \leq \frac{Lp_i^{n-1}}{2(n-1)}, \text{ NEW} < 2L\delta_i\right) \cdot$$
$$\mathbb{E}_{\substack{\tau \sim \text{REVR}(L, \alpha_t, N) \\ \text{SEP}_n \leq \frac{Lp_i^{n-1}}{2(n-1)} \\ \text{NEW} < 2L\delta_i}} \text{DEP}_n(\mathcal{M}_i, \mathcal{M}_{<i}) \quad \textbf{(Case 2)}$$

$$+ \Pr\left(\text{SEP}_n > \frac{Lp_i^{n-1}}{2(n-1)}, \text{ NEW} < 2L\delta_i\right) \cdot$$
$$\mathbb{E}_{\substack{\tau \sim \text{REVR}(L, \alpha_t, N) \\ \text{SEP}_n > \frac{Lp_i^{n-1}}{2(n-1)} \\ \text{NEW} < 2L\delta_i}} \text{DEP}_n(\mathcal{M}_i, \mathcal{M}_{<i}) \quad \textbf{(Case 3)}$$

We estimate these three cases separately.

**Case 1:** $\text{NEW} \geq 2L\delta_i$.

By the definitions of $\text{DEP}_n(\mathcal{M}_i, \mathcal{M}_{<i})$ and NEW, we have:

$$\text{DEP}_n(\mathcal{M}_i, \mathcal{M}_{<i}) \leq |\mathcal{M}_i| = \text{NEW}.$$

Substituting this into the estimation, we obtain:

$$\Pr\left(\text{NEW} \geq 2L\delta_i\right) \cdot \mathbb{E}_{\substack{\tau \sim \text{REVR}(L, \alpha_t, N) \\ \text{NEW} \geq 2L\delta_i}} \text{DEP}_n(\mathcal{M}_i, \mathcal{M}_{<i}) \leq \Pr\left(\text{NEW} \geq 2L\delta_i\right) \cdot \mathbb{E}_{\substack{\tau \sim \text{REVR}(L, \alpha_t, N) \\ \text{NEW} \geq 2L\delta_i}} \text{NEW}$$

Since $\text{DEP}_n(\mathcal{M}_i, \mathcal{M}_{<i}) \geq 0$, the expectation can be expressed as an integral of the tail probability:

$$\mathbb{E}_{\substack{\tau \sim \text{REVR}(L, \alpha_t, N) \\ \text{NEW} \geq 2L\delta_i}} \text{NEW} = \int_{2L\delta_i}^{+\infty} \Pr\left(\text{NEW} \geq x \mid \text{NEW} \geq 2L\delta_i\right) dx.$$

It directly follows that:

$$\Pr\left(\text{NEW} \geq 2L\delta_i\right) \cdot \mathbb{E}_{\substack{\tau \sim \text{REVR}(L, \alpha_t, N) \\ \text{NEW} \geq 2L\delta_i}} \text{NEW} = \Pr\left(\text{NEW} \geq 2L\delta_i\right) \cdot \int_{2L\delta_i}^{+\infty} \Pr\left(\text{NEW} \geq x \mid \text{NEW} \geq 2L\delta_i\right) dx$$

$$= \int_{2L\delta_i}^{+\infty} \Pr\left(\text{NEW} \geq x \mid \text{NEW} \geq 2L\delta_i\right) \Pr\left(\text{NEW} \geq 2L\delta_i\right) dx$$

$$= \int_{2L\delta_i}^{+\infty} \Pr\left(\text{NEW} \geq x\right) dx.$$

Using the same trick as Lemma D.10, applying Lemma C.2, we can derive the bound for probability $\Pr\left(\text{NEW} \geq x\right)$ as:

$$\Pr\left(\text{NEW} \geq x\right) \leq e^{-\frac{(x - L\delta_i)^2}{x + L\delta_i}}.$$

Note that NEW $\leq L$, we only need to consider $2\delta_i \leq 1$. In this case, we have:

$$\int_{2L\delta_i}^{+\infty} \Pr\left(\text{NEW} \geq x\right) dx \leq \int_{2L\delta_i}^{L} e^{-\frac{(x-L\delta_i)^2}{x+L\delta_i}} dx$$

Let $y = x - L\delta_i \in [L\delta_i, L(1-\delta_i)]$, the integral can be rewritten as:

$$\int_{2L\delta_i}^{+\infty} \Pr\left(\text{NEW} \geq x\right) dx \leq \int_{L\delta_i}^{L(1-\delta_i)} e^{-\frac{y^2}{y+2L\delta_i}} dy.$$

Observe that $y + 2L\delta_i \leq 3y$, we can obtain:

$$\int_{2L\delta_i}^{+\infty} \Pr\left(\text{NEW} \geq x\right) dx \leq \int_{L\delta_i}^{L(1-\delta_i)} e^{-\frac{y^2}{3y}} dy = 3\left(e^{-\frac{L\delta_i}{3}} - e^{-\frac{L(1-\delta_i)}{3}}\right) = 3e^{-\frac{L\delta_i}{3}}\left(1 - e^{-\frac{L(1-2\delta_i)}{3}}\right).$$

Using the fact that $e^{-x} \leq \frac{1}{1+x}$ for $x \geq 0$, we have the upper bound:

$$3e^{-\frac{L\delta_i}{3}}\left(1 - e^{-\frac{L(1-2\delta_i)}{3}}\right) \leq 3e^{-\frac{L\delta_i}{3}} \leq \frac{9}{3 + L\delta_i}.$$

Combining the above results, we know that:

$$\Pr\left(\text{NEW} \geq 2L\delta_i\right) \cdot \mathbb{E}_{\substack{\tau \sim \text{REVR}(L,\alpha_t,N) \\ \text{NEW} \geq 2L\delta_i}} \text{DEP}_n(\mathcal{M}_i, \mathcal{M}_{<i}) \leq \frac{9}{3 + L\delta_i}.$$

**Case 2:** $\text{SEP}_n \leq \frac{Lp_i^{n-1}}{2(n-1)}$ and $\text{NEW} < 2L\delta_i$.

Similar to Case 1, we have:

$$\text{DEP}_n(\mathcal{M}_i, \mathcal{M}_{<i}) \leq \text{NEW} < 2L\delta_i,$$

so the expectation also follows:

$$\mathbb{E}_{\substack{\tau \sim \text{REVR}(L,\alpha_t,N) \\ \text{SEP}_n \leq \frac{Lp_i^{n-1}}{2(n-1)} \\ \text{NEW} < 2L\delta_i}} \text{DEP}_n(\mathcal{M}_i, \mathcal{M}_{<i}) < 2L\delta_i.$$

Using the probability bound, it follows that:

$$\Pr\left(\text{SEP}_n \leq \frac{Lp_i^{n-1}}{2(n-1)},\ \text{NEW} < 2L\delta_i\right) \leq \Pr\left(\text{SEP}_n \leq \frac{Lp_i^{n-1}}{2(n-1)}\right) \leq e^{-\frac{Lp_i^{n-1}}{8(n-1)}}.$$

Since $e^{-x} \leq \frac{1}{1+x}$ for $x \geq 0$:

$$e^{-\frac{Lp_i^{n-1}}{8(n-1)}} \leq \frac{8(n-1)}{Lp_i^{n-1} + 8(n-1)}.$$

Combining these results, we obtain:

$$\Pr\left(\text{SEP}_n \leq \frac{Lp_i^{n-1}}{2(n-1)},\ \text{NEW} < 2L\delta_i\right) \cdot \mathbb{E}_{\substack{\tau \sim \text{REVR}(L,\alpha_t,N) \\ \text{SEP}_n \leq \frac{Lp_i^{n-1}}{2(n-1)} \\ \text{NEW} < 2L\delta_i}} \text{DEP}_n(\mathcal{M}_i, \mathcal{M}_{<i}) \leq \frac{16(n-1)L\delta_i}{Lp_i^{n-1} + 8(n-1)}.$$

**Case 3:** $\text{SEP}_n > \frac{Lp_i^{n-1}}{2(n-1)}$ and $\text{NEW} < 2L\delta_i$.

Apparently, we have:

$$\Pr\left(\text{SEP}_n > \frac{Lp_i^{n-1}}{2(n-1)},\ \text{NEW} < 2L\delta_i\right) \leq 1.$$

Given $a, b$, let $\mathbb{E}_{a,b} \text{DEP}_n(\mathcal{M}_i, \mathcal{M}_{<i})$ denote the expectation of $\text{DEP}_n(\mathcal{M}_i, \mathcal{M}_{<i})$ under the condition of $\text{SEP}_n = a$ and $\text{NEW} = b$. In other words:

$$\mathbb{E}_{a,b} \text{DEP}_n(\mathcal{M}_i, \mathcal{M}_{<i}) = \mathbb{E}_{\substack{\tau \sim \text{REVR}(L,\alpha_t,N) \\ \text{SEP}_n = a \\ \text{NEW} = b}} \text{DEP}_n(\mathcal{M}_i, \mathcal{M}_{<i}).$$

Since all the locations are sampled independently, and $\mathrm{DEP}_n(\mathcal{M}_i, \mathcal{M}_{<i})$ depends only on the relative positions of separators in $\mathcal{M}_{<i}$ and the new locations in $\mathcal{M}_i$, the expectation $\mathbb{E}_{a,b}\,\mathrm{DEP}_n(\mathcal{M}_i, \mathcal{M}_{<i})$ only depends on the ordering of separators and new locations.

Assume $x_1, \cdots, x_{a+b}$ are $a + b$ positions (not locations) in order. We can regard the process of ordering separators and new locations as the process of choosing $b$ positions randomly from $x_j$. For $1 \le j \le a + b - 1$, define $X_j = 1$ if $x_j$ and $x_{j+1}$ are both new locations, and $X_j = 0$ otherwise. By the definition of $\mathrm{DEP}_n(\mathcal{M}_i, \mathcal{M}_{<i})$, we can obtain:

$$\mathrm{DEP}_n(\mathcal{M}_i, \mathcal{M}_{<i}) = \sum_{j=1}^{a+b-1} X_j.$$

Since the $b$ new locations are chosen randomly, the probability of $X_j = 1$ can be calculated as:

$$\Pr(X_j = 1) = \frac{C_{a+b-2}^{b-2}}{C_{a+b}^{b}} = \frac{b(b-1)}{(a+b)(a+b-1)}.$$

Therefore, the expectation of $X_j$ is:

$$\mathbb{E}X_j = \frac{b(b-1)}{(a+b)(a+b-1)}.$$

Summing up, we have:

$$\mathbb{E}_{a,b}\,\mathrm{DEP}_n(\mathcal{M}_i, \mathcal{M}_{<i}) = \mathbb{E}\sum_{j=1}^{a+b-1} X_j = (a+b-1)\mathbb{E}X_1 = \frac{b(b-1)}{a+b}.$$

Since $a > \frac{Lp_i^{n-1}}{2(n-1)}$ and $b < 2L\delta_i$, we can derive the upper bound for any $a, b$:

$$\frac{b(b-1)}{a+b} \le \frac{b(b-1)}{\frac{Lp_i^{n-1}}{2(n-1)} + b} \le \frac{2L\delta_i(2L\delta_i - 1)}{\frac{Lp_i^{n-1}}{2(n-1)} + 2L\delta_i} \le \frac{8(n-1)L\delta_i^2}{p_i^{n-1} + 4(n-1)\delta_i}.$$

Since this holds for all $a$ and $b$, we can obtain:

$$\Pr\left(\mathrm{SEP}_n > \frac{Lp_i^{n-1}}{2(n-1)}, \ \mathrm{NEW} < 2L\delta_i\right) \cdot \mathbb{E}_{\substack{\tau\sim\mathrm{REVR}(L,\alpha_t,N) \\ \mathrm{SEP}_n > \frac{Lp_i^{n-1}}{2(n-1)} \\ \mathrm{NEW} < 2L\delta_i}} \mathrm{DEP}_n(\mathcal{M}_i, \mathcal{M}_{<i})$$

$$\le \mathbb{E}_{\substack{\tau\sim\mathrm{REVR}(L,\alpha_t,N) \\ \mathrm{SEP}_n > \frac{Lp_i^{n-1}}{2(n-1)} \\ \mathrm{NEW} < 2L\delta_i}} \mathrm{DEP}_n(\mathcal{M}_i, \mathcal{M}_{<i})$$

$$= \sum_{\substack{a > \frac{Lp_i^{n-1}}{2(n-1)},\ b < 2L\delta_i}} \Pr(\mathrm{SEP}_n = a, \mathrm{NEW} = b) \cdot \mathbb{E}_{a,b}\,\mathrm{DEP}_n(\mathcal{M}_i, \mathcal{M}_{<i})$$

$$\le \frac{8(n-1)L\delta_i^2}{p_i^{n-1} + 4(n-1)\delta_i}.$$

**Summarize the above proof:**

Combining the above three cases, we can obtain:

$$\mathbb{E}_{\tau\sim\mathrm{REVR}(L,\alpha_t,N)}\,\mathrm{DEP}_n(\mathcal{M}_i, \mathcal{M}_{<i}) \le \frac{9}{3 + L\delta_i} + \frac{16(n-1)L\delta_i}{Lp_i^{n-1} + 8(n-1)} + \frac{8(n-1)L\delta_i^2}{p_i^{n-1} + 4(n-1)\delta_i}.$$

If we have the assumption $L\delta_i \ge 1$, it is easy to find that:

$$\mathbb{E}_{\tau\sim\mathrm{REVR}(L,\alpha_t,N)}\,\mathrm{DEP}_n(\mathcal{M}_i, \mathcal{M}_{<i}) \le \frac{9}{3 + L\delta_i} + \frac{16(n-1)\delta_i}{p_i^{n-1}} + \frac{8(n-1)L\delta_i^2}{p_i^{n-1}}$$

$$\le \frac{9}{3 + L\delta_i} + \frac{16(n-1)L\delta_i^2}{p_i^{n-1}} + \frac{8(n-1)L\delta_i^2}{p_i^{n-1}}$$

$$\le \frac{9}{3 + L\delta_i} + \frac{C(n-1)L\delta_i^2}{p_i^{n-1}}.$$

Where $C = 24$ is a constant.

$\square$

Finally, we can derive the upper bound for the KL divergence between the distribution of sequences generated by the reverse model and the ground-truth distribution in the n-gram setting.

**Lemma D.12** (Efficient Sampling with Small KL Divergence). *Let $q$ denote the data distribution over sequences of length $L$, and let $p$ denote the distribution over sequences of length $L$ generated by the reverse model $p_\theta$ with a masking schedule $\alpha_t$ and $N$ reverse steps. Assume that $p_\theta$ satisfies Assumption 4.1. For any $\epsilon > 0$, there exists a masking schedule $\alpha_t$ such that, for $L \geq \frac{3C(n-1)}{\epsilon^{n+\frac{1}{2}}}$, with $N = O\left(\frac{n-1}{\epsilon^n}\right)$ sampling steps, the KL divergence between $q$ and $p$ satisfies:*

$$\frac{D_{\mathrm{KL}}(q\|p)}{L \log |\mathcal{V}|} \leq 4\epsilon + \frac{\epsilon_{\text{learning}}}{\log |\mathcal{V}|}.$$

*Proof.* By Lemma D.9, we know that:

$$D_{\mathrm{KL}}(q\|p) \leq \log |\mathcal{V}| \sum_{i=1}^{N} \mathbb{E}_{\tau \sim \mathrm{REVR}(L,\alpha_t,N)} \mathrm{DEP}_n(\mathcal{M}_i, \mathcal{M}_{<i}) + L\epsilon_{\text{learning}}.$$

Note that at step $t_1$, the reverse process can be bounded using Lemma C.1. By reviewing our proof process, it is easy to see that we can substitute $\mathrm{DEP}_n(\mathcal{M}_1, \mathcal{M}_{<1})$ for $(|\mathcal{M}_1| - 1) \log |\mathcal{V}|$, where $\mathcal{V}$ stands for the vocabulary. By the definition of $\delta_i$, we know that:

$$\mathbb{E}_{\tau \sim \mathrm{REVR}(L,\alpha_t,N)}(|\mathcal{M}_1| - 1) \log |\mathcal{V}| = (\delta_1 L - 1) \log |\mathcal{V}|.$$

Applying Lemma D.11 to $\mathrm{DEP}_n(\mathcal{M}_i, \mathcal{M}_{<i})$, if $L\delta_i \geq 1$, we can obtain:

$$D_{\mathrm{KL}}(q\|p) \leq \delta_1 \log |\mathcal{V}| + \log |\mathcal{V}| \sum_{i=2}^{N} \left( \frac{9}{3 + L\delta_i} + \frac{C(n-1)L\delta_i^2}{p_i^{n-1}} \right) + L\epsilon_{\text{learning}}.$$

By the definition of $p_i$, we know that $p_2 = \delta_1$. For any small $\epsilon > 0$, consider the following masking schedule:

$$\delta_1 = \epsilon, \quad \delta_i = \delta = \frac{\epsilon^n}{C(n-1)}, \quad p_i = \delta_1 + (i-2)\delta, \quad \forall i \geq 2.$$

Then, for $L \geq \frac{1}{\delta}$, the KL divergence can be bounded by:

$$\frac{D_{\mathrm{KL}}(q\|p)}{L \log |\mathcal{V}|} \leq \epsilon + \frac{9(N-1)}{L(3 + L\delta)} + \sum_{i=2}^{N} \frac{C(n-1)\delta^2}{p_i^{n-1}} + \frac{\epsilon_{\text{learning}}}{\log |\mathcal{V}|}$$

$$= \epsilon + \frac{9(1 - \delta_1)}{L\delta(3 + L\delta)} + \frac{C(n-1)\delta^2}{\delta_1^{n-1}} + \sum_{i=1}^{N-2} \frac{C(n-1)\delta^2}{(\delta_1 + i\delta)^{n-1}} + \frac{\epsilon_{\text{learning}}}{\log |\mathcal{V}|}.$$

$$\leq \epsilon + \frac{9}{L\delta(3 + L\delta)} + \frac{C(n-1)\delta^2}{\delta_1^{n-1}} + \sum_{i=1}^{N-2} \frac{C(n-1)\delta^2}{(\delta_1 + i\delta)^n} + \frac{\epsilon_{\text{learning}}}{\log |\mathcal{V}|}.$$

By simple calculations, we know that:

$$\frac{9}{L\delta(3 + L\delta)} \leq \epsilon, \quad \text{if } L \geq \frac{3}{\delta\epsilon^{\frac{1}{2}}}.$$

It is clear that $\delta \leq 1$, so:

$$\frac{C(n-1)\delta^2}{\delta_1^{n-1}} \leq \epsilon\delta \leq \epsilon.$$

Since $x^{-n}$ is convex on $[0, +\infty)$, the accumulation can be bounded by:

$$\sum_{i=1}^{N-2} \frac{C(n-1)\delta^2}{(\delta_1 + i\delta)^n} = C(n-1)\delta^{2-n} \sum_{i=1}^{N-2} \frac{1}{(\frac{\delta_1}{\delta} + i)^n}$$

$$\leq C(n-1)\delta^{2-n} \sum_{i=1}^{N-2} \int_{x=0}^{+\infty} \frac{1}{(\frac{\delta_1}{\delta} + x)^n} \mathrm{d}x$$

$$= C(n-1)\delta^{2-n} \cdot \frac{1}{n-1} \left(\frac{\delta}{\delta_1}\right)^{n-1}$$

$$= \frac{C\delta}{\delta_1^{n-1}}$$

$$\leq \epsilon.$$

Combining the above, we have:

$$\frac{D_{\mathrm{KL}}(q\|p)}{L \log |\mathcal{V}|} \leq 4\epsilon + \frac{\epsilon_{\mathrm{learning}}}{\log |\mathcal{V}|}.$$

Meanwhile, the time step is limited by:

$$N = 1 + \frac{1 - \delta_1}{\delta} = O\left(\frac{n-1}{\epsilon^n}\right),$$

and the lower bound for $L$:

$$L \geq \frac{3}{\delta \epsilon^{\frac{1}{2}}} = \frac{3C(n-1)}{\epsilon^{n+\frac{1}{2}}}.$$

$\square$

Combining the above lemmas, we can prove Theorem 4.2 by breaking the expression of $\log \mathrm{TER}(p)$ into two parts.

**Theorem D.13** (TER Bounds for $n$-Gram Language Generation). *For any $n$-gram language $q$ and any $\epsilon > 0$, let $p_\theta$ denote the reverse model and $L$ denote the sequence length. The distribution over sequences generated by $p_\theta$ is denoted as $p$. For any $L > O\left(\frac{n-1}{\epsilon^{n+0.5}}\right)$, under Assumption 4.1, there exists a masking schedule $\alpha_t$ such that, with $N = O\left(\frac{n-1}{\epsilon^n}\right)$ sampling steps, the perplexity of the MDM is upper-bounded by:*

$$\log \mathrm{TER}(p) \leq \log \mathrm{TER}(q) + \epsilon_{learning} + 4\epsilon \log |\mathcal{V}|.$$

*Proof.* By Lemma D.12, for any $L > O\left(\frac{n-1}{\epsilon^{n+0.5}}\right)$, there exists a masking schedule $\alpha_t$ with $N = O\left(\frac{n-1}{\epsilon^n}\right)$ sampling steps satisfying:

$$\frac{D_{\mathrm{KL}}(q\|p)}{L \log |\mathcal{V}|} \leq 4\epsilon + \frac{\epsilon_{\mathrm{learning}}}{\log |\mathcal{V}|}.$$

In other words:

$$\frac{1}{L} \mathbb{E}_{\boldsymbol{x} \sim q} \log \frac{q(\boldsymbol{x})}{p(\boldsymbol{x})} \leq 4\epsilon \log |\mathcal{V}| + \epsilon_{\mathrm{learning}}.$$

By the definition of TER, we have:

$$\log \mathrm{TER}(p) = \mathbb{E}_{\boldsymbol{x} \sim q} - \frac{\log p(\boldsymbol{x})}{|\boldsymbol{x}|} = \frac{1}{L} \mathbb{E}_{\boldsymbol{x} \sim q} \left(-\log q(\boldsymbol{x}) + \log \frac{q(\boldsymbol{x})}{p(\boldsymbol{x})}\right).$$

Note that:

$$\log \mathrm{TER}(q) = \mathbb{E}_{\boldsymbol{x} \sim q} - \frac{\log q(\boldsymbol{x})}{|\boldsymbol{x}|} = \frac{1}{L} \mathbb{E}_{\boldsymbol{x} \sim q} - \log q(\boldsymbol{x}).$$

We can obtain:

$$\log \mathrm{TER}(p) \leq \log \mathrm{TER}(q) + \epsilon_{\mathrm{learning}} + 4\epsilon \log |\mathcal{V}|.$$

$\square$

# E   Proof for Theorem 4.3 and Theorem 4.4

## E.1   Proof for Theorem 4.3

In this section, we aim to derive the upper bound for the SER of generated sequences with sufficient reverse steps. First, we argue that, given a making schedule $\alpha_t$, with sufficient steps, the probability of sampling multiple locations in the sequence at the same time can be very low.

**Lemma E.1** (Low Probability of Simultaneous Sampling with Sufficient Steps). *Given a sequence of length L and a masking schedule $\alpha_t$. For any $\epsilon > 0$, there exists $N_0$, such that for any $N \geq N_0$, with N reverse steps, the probability $p_{\mathrm{mul}}$ of sampling multiple locations in the sequence at the same time satisfies:*

$$p_{\mathrm{mul}} < \epsilon.$$

*Proof.* By Lemma D.10, we know that the probability of a location being sampled at time step $t_i = \frac{N-i}{N}$ is:

$$\delta_i = \alpha_{t_i} - \alpha_{t_{i-1}} = \alpha_{\frac{N-i}{N}} - \alpha_{\frac{N-i+1}{N}}.$$

Since all the locations are sampled independently, for two distinct locations $i \neq j$ in the sequence, the probability that $i$ and $j$ are sampled simultaneously is:

$$p_{i,j} = \sum_{i=1}^{N} \delta_i^2.$$

Summing up $p_{i,j}$, the probability of having two locations a=sampled simultaneously can be bounded by:

$$p_{\mathrm{mul}} \leq \frac{L(L-1)}{2} \cdot \sum_{i=1}^{N} \delta_i^2$$

Since $\alpha_t$ is continuous on $[0,1]$, we know that it is uniformly continuous. Therefore, for any $\epsilon > 0$, there exists $N_0 > 0$ that satisfies:

$$|\alpha_x - \alpha_y| < \frac{2\epsilon}{L(L-1)}, \quad \forall x, y \in [0,1], |x-y| < \frac{1}{N_0}.$$

In this case, for $N > N_0$, we know that:

$$|\delta_i| = |\alpha_{\frac{N-i}{N}} - \alpha_{\frac{N-i+1}{N}}| < \frac{2\epsilon}{L(L-1)}, \quad \forall i \in [N].$$

Combining with the fact that $\sum_{i=1}^{N} \delta_i = 1$, we can obtain:

$$p_{\mathrm{mul}} \leq \frac{L(L-1)}{2} \cdot \sum_{i=1}^{N} \delta_i \cdot \max_{j \in [N]} \delta_j < \epsilon.$$

$\square$

Next, we consider the SER increase due to the learning error. Specifically, we only investigate the case where all the locations are sampled at different steps.

**Lemma E.2** (Accurate Step-by-Step Generation with Low Learning Error). *Let q denote any HMM, and let $p_\theta$ represent the reverse model under an arbitrary masking schedule, where L is the sequence length. Let p denote the distribution over sequences generated by $p_\theta$. Under Assumption 4.1 with a learning error $\epsilon_{learning} < \frac{\delta}{L}$, $\delta > 0$, and given an instance of reverse process $\tau = (\mathcal{M}_1, \mathcal{M}_2, \cdots, \mathcal{M}_N)$ with $|\mathcal{M}_i| \leq 1$, let $p_{\mathrm{acc}}$ denote the probability of generating a valid sequence. Then $p_{\mathrm{acc}}$ satisfies:*

$$p_{\mathrm{acc}} \geq e^{-\delta}.$$

*Proof.* Since $|\mathcal{M}_i| \leq 1$, we only need to consider the steps where one token is sampled. Let $\tilde{x}_t$ denote the previously sampled tokens, and $\tilde{x}_t$ denote the token sampled at the current step. If $\tilde{x}_t$

is can later form a valid sequence, let $\mathcal{X}_t$ denote the set of valid choices for $\tilde{x}_t$. In other words, if $\tilde{x}_t \in \mathcal{X}_t$, then the combination of $\tilde{\boldsymbol{x}}_t$ and $\tilde{x}_t$ is can later form a valid sequence, or more intuitively:

$$q_{0|t}(\tilde{x}_t \mid \tilde{\boldsymbol{x}}_t) > 0.$$

Under Assumption 4.1, we know that:

$$D_{\mathrm{KL}}(q_{0|t}(x_t \mid \tilde{\boldsymbol{x}}_t) \| p_\theta(x_t \mid \tilde{\boldsymbol{x}}_t)) < \epsilon_{\text{learning}}.$$

Since it is assumed that $0 \log 0 = 0$, we have:

$$\sum_{x_t \in \mathcal{X}_t} q_{0|t}(x_t \mid \tilde{\boldsymbol{x}}_t) \log \frac{q_{0|t}(x_t \mid \tilde{\boldsymbol{x}}_t)}{p_\theta(x_t \mid \tilde{\boldsymbol{x}}_t)} < \epsilon_{\text{learning}}.$$

Equivalently, we have:

$$-\epsilon_{\text{learning}} < \sum_{x_t \in \mathcal{X}_t} q_{0|t}(x_t \mid \tilde{\boldsymbol{x}}_t) \log \frac{p_\theta(x_t \mid \tilde{\boldsymbol{x}}_t)}{q_{0|t}(x_t \mid \tilde{\boldsymbol{x}}_t)}.$$

Due to the concavity of $\log x$, by Jensen's Inequality, we can obtain:

$$\sum_{x_t \in \mathcal{X}_t} q_{0|t}(x_t \mid \tilde{\boldsymbol{x}}_t) \log \frac{p_\theta(x_t \mid \tilde{\boldsymbol{x}}_t)}{q_{0|t}(x_t \mid \tilde{\boldsymbol{x}}_t)} \leq \log \left( \sum_{x_t \in \mathcal{X}_t} q_{0|t}(x_t \mid \tilde{\boldsymbol{x}}_t) \cdot \frac{p_\theta(x_t \mid \tilde{\boldsymbol{x}}_t)}{q_{0|t}(x_t \mid \tilde{\boldsymbol{x}}_t)} \right) = \log \sum_{x_t \in \mathcal{X}_t} p_\theta(x_t \mid \tilde{\boldsymbol{x}}_t).$$

Therefore, the probability that each step remains valid satisfies:

$$\sum_{x_t \in \mathcal{X}_t} p_\theta(x_t \mid \tilde{\boldsymbol{x}}_t) \geq e^{-\epsilon_{\text{learning}}} \geq e^{-\frac{\delta}{L}}.$$

Since there are $L$ locations in the sequence, the probability of generating a valid sequence is bounded by:

$$p_{\text{acc}} \geq (e^{-\frac{\delta}{L}})^L = e^{-\delta}.$$

$\square$

Combining the above lemmas, we can derive the upper bound of SER by taking sufficient reverse steps and small learning error.

**Theorem E.3** (Accurate Generation of HMM with Sufficient Steps). *Let $q$ denote any HMM, and let $p_\theta$ represent the reverse model under an arbitrary masking schedule, where $L$ is the sequence length. Let $p$ denote the distribution over sequences generated by $p_\theta$. Under Assumption 4.1 with a learning error $\epsilon_{learning} < O(\frac{\delta}{L})$, and given a sufficient number of reverse steps, the sequence error rate $\mathrm{SER}(p)$ of the generated text satisfies*

$$\mathrm{SER}(p) \leq \delta.$$

*Proof.* For $\delta > 0$, we know that:
$$1 - \delta < c.$$

By Lemma E.1, given the masking schedule $\alpha_t$, there exists $N_0$, for $N > N_0$ and $N$ reverse steps, the probability of sampling multiple locations in the sequence at the same time is bounded by:

$$p_{\text{mul}} < 1 - \frac{1 - \delta}{e^{-\delta}}.$$

In other words, the probability of sampling all the locations at different steps is at least $\frac{1-\delta}{e^{-\delta}}$. By Lemma E.2, for each reverse process which satisfies that all the locations are sampled at different steps, the probability of generating a valid sequence is lower bounded by:

$$p_{\text{acc}} \geq e^{-\delta}.$$

Therefore, the sequence error rate SER satisfies:

$$\mathrm{SER}(p) \leq 1 - \frac{1 - \delta}{e^{-\delta}} \cdot e^{-\delta} = \delta.$$

$\square$

## E.2 Proof for Theorem 4.4

In the section, we aim to find an example (Example E.7) with high sequence error rate. To present this example, we begin with a special class of languages defined under the interval setting:

**Definition E.4** (Interval Setting)**.** Consider a sequence of length $L$, which is divided equally into $M$ intervals $\mathcal{I}_1, \mathcal{I}_2, \cdots, \mathcal{I}_M$, each of length $l = \frac{L}{M} \geq 2$. Given a masking schedule $\alpha_t$, an instance of reverse process $\tau = (\mathcal{M}_1, \mathcal{M}_2, \cdots, \mathcal{M}_N)$ is defined by Definition D.1. For any two locations within different intervals, their corresponding tokens are independent from each other. In other words, let $\tilde{\boldsymbol{x}}_i^{(j)}$ denote the new tokens in $\mathcal{M}_i \cap \mathcal{I}_j$, $\tilde{\boldsymbol{x}}_{<i}^{(j)}$ denote the previously sampled tokens in $\mathcal{M}_{<i} \cap \mathcal{I}_j$, and $p$ denote the distribution over sequences generated by the reverse model with reverse process $\tau$, then for time step $t_i = \frac{N-i}{N}$:

$$p(\tilde{\boldsymbol{x}}_i^{(j)} | \tilde{\boldsymbol{x}}_{<i}) = p(\tilde{\boldsymbol{x}}_i^{(j)} | \tilde{\boldsymbol{x}}_{<i}^{(j)}).$$

In this case, we have:

$$p(\boldsymbol{x}) = \prod_{j=1}^{M} p(\boldsymbol{x}^{(j)}) = \prod_{j=1}^{M} \prod_{i=1}^{N} p(\tilde{\boldsymbol{x}}_i^{(j)} | \tilde{\boldsymbol{x}}_{<i}^{(j)}).$$

We denote the above setting as $\text{Inter}(L, l, \alpha_t)$.

Under the interval setting defined above, we can control the probability of sampling simultaneously in the same interval.

**Lemma E.5** (Simultaneous Sampling Probability for an Interval)**.** *Consider the interval setting* $\text{Inter}(L, l, \alpha_t)$*. For each interval* $\mathcal{I}_j$ *of length* $l$*, let* $h_j$ *denote the probability that all the locations in* $\mathcal{I}_j$ *are sampled in different time steps. Then,* $h_j$ *can be bounded by:*

$$h_j \leq 1 - \frac{1}{N}.$$

*Proof.* Let $\delta_i = \alpha_{t_i} - \alpha_{t_{i-1}}$. Similar to Lemma D.10, we know that $\delta_i$ is the probability of a location being sampled at time step $t_i$. Take the first location in $|\mathcal{I}_j|$, denote it as $X_1$, and let $X_2, \cdots, X_l$ denote the rest $l - 1$ locations in $\mathcal{I}_j$. If $X_1$ is sampled at step $t_i$, then $X_2, \cdots, X_l$ must be sampled at time steps other than $t_i$. Therefore, $h_j$ can be bounded by:

$$h_j \leq \sum_{i=1}^{N} \delta_i (1 - \delta_i)^{l-1} \leq \sum_{i=1}^{N} \delta_i (1 - \delta_i).$$

Let $f(\delta) = \delta(1 - \delta)$. Note that we have:

$$f''(\delta) = -2 \leq 0,$$

which indicates that $f(\delta)$ is concave. Using Jensen's Inequality, we can obtain:

$$h_j \leq \sum_{i=1}^{N} f(\delta_i) \leq N f\left(\frac{1}{N}\right) = 1 - \frac{1}{N}.$$

$\square$

Using the above lemma, if we assume that sampling simultaneously in one interval increases SER, then we can derive an lower bound for $\text{SER}(p)$.

**Lemma E.6** (SER bound for Interval Setting)**.** *Consider the interval setting* $\text{Inter}(L, l, \alpha_t)$*. Assume that sampling simultaneously in the same interval introduces an error with probability at least* $p_0$*, and other actions do not reduce error. In other words, if two locations in an interval are both sampled at step* $t_i$*, then there is a probability of* $p_e$ *that the sequence will not be accurate afterwards. In this case, let* $p$ *denote the distribution over sequences of length* $L$ *generated by the reverse model with masking schedule* $\alpha_t$ *and* $N$ *reverse steps. We have the following bound for* SER*:*

$$\text{SER}(p) \geq 1 - \left(1 - \frac{p_e}{N}\right)^{L/l}.$$

*Proof.* By Lemma E.5, we can obtain that for each interval $\mathcal{I}_j$, the probability $p_{\text{error}}^{(j)}$ of generating an error in $\mathcal{I}_j$ is lower-bounded by:

$$p_{\text{error}}^{(j)} \geq p_e(1 - h_j) \geq \frac{p_e}{N}.$$

Due to the independence between different intervals, the accuracy $\text{SER}(p)$ can be calculated as:

$$\text{SER}(p) = 1 - \prod_{j=1}^{M}(1 - p_{\text{error}}^{(j)}).$$

Therefore, we have the bound:

$$\text{SER}(p) \geq 1 - \left(1 - \frac{p_e}{N}\right)^{L/l}.$$

$\square$

To show that the above setting is reasonable and achievable, we give the following example, which is later shown to be the example we are looking for.

**Example E.7.** Consider a sequence of length $L$, which is divided equally into $M$ intervals, each of length $l = L/M$. Denote the $k$-th interval as $\mathcal{I}_k = [1 + (k-1)l, \, kl]$. The tokens $x_i$, $1 \leq i \leq L$ in the sequence satisfy the following rules:

- Each $x_i$ takes values in the set $\mathcal{A} = \{a_1, \cdots, a_{2^{l-1}}\}$. For each $a_j \in \mathcal{A}$, there corresponds a vector $v_j = (v_{j,1}, \cdots, v_{j,l-1}) \in \{0,1\}^{l-1}$, where $(v_{j,1} \cdots v_{j,l-1})_2$ is the binary expression for $j - 1$. Thus, each random variable $x_i$ corresponds to a random vector $(v_1^{(i)}, \cdots, v_{l-1}^{(i)})$, where $v_j^{(i)} \in \{0,1\}$ for $j = 1, \cdots l - 1$.

- For $i \in \mathcal{I}_k$ and $j \in \mathcal{I}_s$, if $k \neq s$, then $x_i$ and $x_j$ are independent.

- For $i, j \in \mathcal{I}_k$ such that $i < j$, let $i' = i - (s-1)l$ and $j' = j - (s-1)l$. Then, $x_i$ and $x_j$ are the $i'$-th and $j'$-th elements in interval $\mathcal{I}_k$, respectively. The corresponding binary components satisfy $v_{j'-1}^{(i)} = v_{i'}^{(j)} \sim \text{Bernoulli}(\frac{1}{2})$, which is independent of all other $v_t^{(s)}$.

In this setup, each interval $\mathcal{I}_k$ contains $\frac{l(l-1)}{2}$ pairs of mutually independent random variables. Given an arbitrary masking schedule $\alpha_t$, this setting is consistent with Definition E.4. Let $q$ denote the data distribution described above.

Under Assumption 4.1, we only need to examine the case where $\boldsymbol{x}_t$ has no error. By Lemma C.3, we know that:

$$\left\|q_{0|t}(x_0^i \mid \boldsymbol{x}_t) - p_\theta(x_0^i \mid \boldsymbol{x}_t)\right\|_1 \leq \sqrt{2 D_{\text{KL}}(q_{0|t}(x_0^i \mid \boldsymbol{x}_t) \| p_\theta(x_0^i \mid \boldsymbol{x}_t))} \leq \sqrt{2\epsilon_{learning}}.$$

Let $\mathcal{M}$ denote the set of previously sampled locations. For $q$ and any unsampled location in interval $\mathcal{I}$, all of the potential tokens $x$ at this location which is consistent with $\boldsymbol{x}_t$ have the same probability:

$$q(x|\boldsymbol{x}_t) = \frac{1}{2^{l-1-|\mathcal{M}\cap\mathcal{I}|}}.$$

If two locations $x_i, x_j$ within the same interval $\mathcal{I}$ are sampled simultaneously, ignoring the possible inconsistency with previously sampled tokens (since error can not be reduced), the independence of the random variable pairs implies that the probability of generating an error is lower-bounded by:

$$p_e \geq (\frac{1}{2} + e_1)(\frac{1}{2} + e_2) + (\frac{1}{2} + e_3)(\frac{1}{2} + e_4)$$

where $\frac{1}{2}$ implies the probability (for $q$) of letting $v_{i'}^{(j)}$ or $v_{j'-1}^{(i)}$ to be 0 or 1, and $e_1, e_2, e_3, e_4$ satisfies:

$$|e_1| + |e_3| = \left\|q_{0|t}(x_0^i \mid \boldsymbol{x}_t) - p_\theta(x_0^i \mid \boldsymbol{x}_t)\right\|_1$$
$$|e_2| + |e_4| = \left\|q_{0|t}(x_0^j \mid \boldsymbol{x}_t) - p_\theta(x_0^j \mid \boldsymbol{x}_t)\right\|_1$$

Thus, we know that:

$$p_e \geq \frac{1}{2} - (|e_1| + |e_2| + |e_3| + |e_4|) \geq \frac{1}{2} - 2\sqrt{2\epsilon_{learning}}.$$

In other words, this is consistent with the setting Lemma E.6, with an error probability $p_e = \frac{1}{2} - 2\sqrt{2\epsilon_{learning}}$.

Although the example above seems a bit tricky, it can actually be modified into the form of an HMM, a commonly considered structure for generative models.

**Note E.8** (HMM Form of Example E.7). The setting described in Example E.7 can be alternatively modeled as a Hidden Markov Model (HMM), where the observation space is $\mathcal{O} = \mathcal{A}$, and the state space is $\mathcal{S} = \{(i, A^{(i)})|A^{(i)} \in \mathbb{R}^{(l-1)\times(l-1)}, i = 1, \cdots, l\}$. Here, $i$ represents the current position within the interval, and $A^{(i)}$ is an upper triangular matrix with entries taking values of 0 or 1. For $j \leq i$, the $j$-th row of $A^{(i)}$ encodes the values sampled by the variable pairs formed between the $j$-th position and all its subsequent positions in the interval. For $j > i$, the $j$-th row of $A^{(i)}$ is set to 0.

Given the current state $s = (i, A^{(i)})$, the state transition and emission process can be describe as follows:

- The observation $o_i$ corresponds to the $i - 1$-th column and the $i$-th row of the matrix $A^{(i)}$, where the values of variable pairs relevant to the $i$-th position within the interval are encoded. Specifically, we know that $o_i \in \mathcal{A}$ corresponds to a vector $v_i = (v_{i,1}, \cdots, v_{i,l-1})$, where
$$v_{i,j} = \begin{cases} A^{(i)}_{j,i-1}, & j < i, \\ A^{(i)}_{i,j}, & j \geq i. \end{cases}$$

- If $i < l$, the next state is $s' = (i, A^{(i+1)})$, where the first $i$ rows of $A^{(i+1)}$ is the same as $A^{(i)}$, and $A^{(i+1)}_{i+1,j} \sim \text{Bernoulli}(\frac{1}{2})$ i.i.d. for $j = i+1, \cdots, l-1$, with the remaining entries set to 0.

- If $i = l$, the next state resets to $s' = (1, A^{(1)})$, where the entries in the first row are independently sampled from $\text{Bernoulli}(\frac{1}{2})$, and other entries are set to 0.

The size of the observation space is given by $|\mathcal{O}| = |\mathcal{A}| = 2^{l-1}$. The size of the state space is computed as:

$$|\mathcal{S}| = \sum_{i=1}^{l} 2^{(2l-i-1)i/2} \leq l \cdot 2^{l(l-1)/2}.$$

The above Note gives the HMM form of Example E.7. In fact, with appropriate adjustments, it can be further modified into an n-gram language. Using the HMM defined above, we can prove Theorem 4.4.

**Theorem E.9** (SER Bound for HMM Generation). *There exists an HMM $q$ over a vocabulary of size 16 that satisfies the following conditions: for any reverse model $p_\theta$ under Assumption 4.1 with $\epsilon_{\text{learning}} < \frac{1}{128}$, and any masking schedule $\alpha_t$, let $p$ denote the distribution over sequences generated by $p_\theta$. There exists a constant $C$ such that if the number of sampling steps satisfies $N = CL$, where $L$ is the sequence length, the SER of the generated text is lower-bounded by:*

$$\text{SER}(p) > \frac{1}{2}.$$

*Proof.* Take the HMM described in Note E.8, and set $l = 5$, $N = CL$. The vocabulary is the observation space $\mathcal{O}$ which satisfies $|\mathcal{O}| = 2^{l-1}$. By Lemma E.6, for any masking schedule $\alpha_t$, we have:

$$\text{SER}(p) \geq 1 - \left(1 - \frac{p_e}{N}\right)^{L/l}.$$

As illustrated in Example E.7:

$$p_e = \frac{1}{2} - 2\sqrt{2\epsilon_{learning}}.$$

Therefore, take $N = CL$, and let $y = \frac{CL}{p_e}$, we have:

$$\text{SER}(p) \geq 1 - \left[\left(1 - \frac{1}{y}\right)^y\right]^{\frac{p_e}{Cl}}.$$

Since $(1 - \frac{1}{y})^y$ is decreasing, and apparently $y \geq \frac{Cl}{p_e}$, we know that:

$$\text{SER}(p) \geq \frac{p_e}{Cl}.$$

Let $C = \frac{2p_e}{l+1}$, we can get the upper bound:

$$\text{SER}(p) > \frac{1}{2}.$$

In this way:

$$C = \frac{2p_e}{l+1} = \frac{\frac{1}{2} - 2\sqrt{2\epsilon_{learning}}}{6} \geq \frac{1}{24} = O(1).$$

$\square$

### E.3 Extending Theorem 4.4 to Remasking Strategies

In this section, we extend the conclusion of Theorem 4.4 to masked diffusion models with remasking sampler (named ReMDM) proposed by (Wang et al., 2025). We begin by presenting the main mechanisms of the sampler, followed by the justification for analogous results.

As is introduced in the paper, ReMDM is an MDM with remasking designs, where preciously sampled tokens have a chance to be remasked and resampled again in later steps. Formally, let $\sigma_t$ be the remasking schedule that satisfies $0 \leq \sigma_t \leq \min\{1, \frac{1-\alpha_s}{\alpha_t}\}$ for any time step $s < t$.

In contrast to the conventional reverse process defined in Equation (2), the reverse process of ReMDM (with the original sequence $x$) is defined as:

$$q_{s|t}(\boldsymbol{x}_s|\boldsymbol{x}_t, \boldsymbol{x}) = \prod_{i=0}^{L-1} q_{s|t}(x_s^i|\boldsymbol{x}_t, \boldsymbol{x}), \quad \text{where}$$

$$q_{s|t}(x_s^i|\boldsymbol{x}_t, \boldsymbol{x}) = \begin{cases} 1 - \sigma_t, & x_t^i \neq \texttt{[m]}, x_s^i = x_t^i, \\ \sigma_t, & x_t^i \neq \texttt{[m]}, x_s^i = \texttt{[m]}, \\ \frac{1-\alpha_s-\sigma_t\alpha_t}{1-\alpha_t}, & x_t^i = \texttt{[m]}, x_s^i = \texttt{[m]}, \\ \frac{\alpha_s-(1-\sigma_t)\alpha_t}{1-\alpha_t}q_{0|t}(x_s^i|\boldsymbol{x}_t), & x_t^i = \texttt{[m]}, x_s^i \neq \texttt{[m]}, \\ 0, & \text{otherwise.} \end{cases} \quad (7)$$

Intuitively, compared to the original reverse process, there is a probability that already generated tokens are remasked during the generation. Now, consider the last time $t$ that a location is sampled. In other words, the location is sampled at step $t$, and is not remasked in later steps, thus the token sampled at $t$ will stay unchanged. We will derive the probability that a location is sampled for the *last time* at time step $t_i = \frac{N-i}{N}$.

**Lemma E.10** (Last Sample Probability Estimate of ReMDM). *Given a masking schedule $\alpha_t$, a remasking schedule $\sigma_t$ and $N$ reverse steps, for ReMDM, the probability $\delta_i$ that a location is sampled for the last time at time step $t_i = \frac{N-i}{N}$ satisfies that:*

$$\delta_i = (\alpha_{t_i} - (1-\sigma_{t_{i-1}})\alpha_{t_{t-1}}) \prod_{j=i}^{N-1}(1-\sigma_{t_j}).$$

*Proof.* It follows from the symmetry that we can consider any location in the sequence. First, we use induction to show the probability $p_m(i)$ that the token at this location is [m] at $t_i$ satisfies $p_m(i) = 1 - \alpha_{t_i}$.

For $i = 0$, since $\alpha_1 = 0$, it is direct that $p_m(0) = 1 = 1 - \alpha_{t_0}$.

For $i = k + 1$, assume that $p_m(k) = 1 - \alpha_{t_k}$. Combining with Equation (7), we know that

$$p_m(k+1) = p_m(k) \cdot \frac{1 - \alpha_{t_{k+1}} - \sigma_{t_k} \alpha_{t_k}}{1 - \alpha_{t_k}} + (1 - p_m(k)) \cdot \sigma_{t_k} = 1 - \alpha_{t_{k+1}}.$$

Thus, we can conclude that $p_m(i) = 1 - \alpha_{t_i}$. Therefore, the probability $\delta_i$ can be decomposed into

$$
\begin{aligned}
\delta_i &= \Pr(\,[\text{m}]\,\text{at}\,t_{t_{i-1}}) \cdot \Pr(\text{not}\,[\text{m}]\,\text{at}\,t_{t_i} \mid [\text{m}]\,\text{at}\,t_{t_{i-1}}) \cdot \Pr(\text{stay unmasked} \mid \text{not}\,[\text{m}]\,\text{at}\,t_{t_i}) \\
&= p_m(i-1) \cdot \frac{\alpha_{t_i} - (1 - \sigma_{t_{i-1}})\alpha_{t_{i-1}}}{1 - \alpha_{t_{i-1}}} \cdot \prod_{j=i}^{N-1} (1 - \sigma_{t_j}) \\
&= (\alpha_{t_i} - (1 - \sigma_{t_{i-1}})\alpha_{t_{t-1}}) \prod_{j=i}^{N-1} (1 - \sigma_{t_j}).
\end{aligned}
$$

$\square$

By replacing the probability of a location being sampled at time step $t$ with the probability of that a location being sampled for the *last time* to be at time step $t$ (and letting $\delta_i$ to be the corresponding probability in Lemma E.10), we can derive analogous results to those presented in Lemma E.5, Lemma E.6, and Theorem E.9. This substitution is justified because when two locations are sampled simultaneously and are not later remasked and resampled, there exists a probability of introducing errors. Therefore, by applying similar proof techniques, we arrive at the same conclusion as stated in Theorem 4.4.

# F Experiment Details

In this section, we will present the details of the experiments.

## F.1 Data Generation

We evaluate the MDMs in a variety of formal languages, including $n$-gram languages and HMMs. For each formal language, parameters are generated through random sampling, we present the sampling algorithm in Algorithm 1 and Algorithm 2. It is notable that to add some deterministic to the language model in the evaluation of SER, we add the parameter of thres to prune the tail probabilities, making sure the language model only generates the correct sequence. For the evaluation of TER, we set the thres to be 0, for the well definition of generative perplexity. The detailed parameters to generate the formal languages are listed in Table 6.

Table 6: Generation Parameters for Different Language Models

| Parameter | 2-gram | 3-gram | 4-gram | HMM |
|---|---|---|---|---|
| vocabulary size | 8 | 8 | 8 | 8 |
| Hidden States ($n$) | N/A | N/A | N/A | 32 |
| Temperature | 2 | 2 | 2 | 3.2 |
| Threshold | 0.008 | 0.008 | 0.005 | 0.003 |

---

**Algorithm 1** Generate $n$-gram Language Model

---

**Input**:
$n$: number of grams
vocab_size: size of vocabulary
temp: temperature (controls randomness, higher indicates more randomness)
thres: threshold for pruning small probabilities
**Output**: $n$-gram language model with parameters:
$T$: transition probability matrix (vocab_size$^{n-1}$ × vocab_size)
Init_dist: initial state distribution

1: Init_dist ← rand(hidden_states_num)
2: Init_dist ← Init_dist/$\sum$(Init_dist)
3: $T$ ← randn(vocab_size$^{n-1}$, vocab_size) × randomness
4: $T$ ← softmax($T$)
5: **if** thres > 0 **then**
6:    $T$[where($T$ < thres)] ← 0
7:    $T$ ← $T$/rowsum($T$)
8: **end if**
9: **return** $T$ and Init_dist

---

---

**Algorithm 2** Generate Hidden Markov Model

---

**Input**:
$n$: number of hidden states
vocab_size: size of vocabulary
randomness: temperature parameter to control probability distributions
thres: threshold for pruning small transition probabilities
**Output**: HMM with parameters:
$A$: state transition matrix ($n \times n$)
$B$: emission probability matrix ($n \times$ (vocab_size + 1))
Init_dist: initial state distribution ($n$-dimensional)

1: hidden_states_num ← $n$
2: Init_dist ← rand(hidden_states_num)
3: Init_dist ← Init_dist/$\sum$(Init_dist)
4: $A$ ← randn(hidden_states_num, hidden_states_num) × randomness
5: $A$ ← softmax($A$)
6: **if** thres > 0 **then**
7:    $A$[where($A$ < thres)] ← 0
8:    $A$ ← $A$/rowsum($A$)
9: **end if**
10: $B$ ← randn(hidden_states_num, vocab_size) × randomness × 2.5
11: $B$ ← softmax($B$)
12: $B$[where($B$ < 0.05)] ← 0
13: $B$ ← $B$/rowsum($B$)
14: $B$ ← concat($B$, ones(hidden_states_num, 1)/hidden_states_num)
15: **return** $A, B$, and Init_dist

---

### F.2 Model Training and Testing

In our experiments of formal languages, all training was conducted on NVIDIA A100 GPUs. The model architectures and train configurations are listed in Table 7 and Table 8. The training configuration of the auto-regressive model is listed in Table 9. We run the each experiment for 5 times and report the mean and standard deviation.

| Model Configuration | |
| --- | --- |
| Hidden Size | 768 |
| Sequence Length | {512,1024,2048} |
| Number of Layers | 10 |
| Attention Heads | 12 |

Table 7: Model Configuration for the Formal Language Tasks

| Training Configuration for MDMs | |
| --- | --- |
| Epochs | 20 |
| Learning Rate | 3e-4 |
| Optimizer | AdamW |
| $\beta_1$ | 0.9 |
| $\beta_2$ | 0.999 |
| Learning Rate Scheduler | Cosine Scheduler with Warmup |
| Warmup Ratio | 0.1 |

Table 8: Training Configuration for MDMs on the Formal Language Tasks

| Training Configuration for Auto-regressive Models | |
| --- | --- |
| Epochs | 20 |
| Learning Rate | 3e-4 |
| Optimizer | AdamW |
| $\beta_1$ | 0.9 |
| $\beta_2$ | 0.999 |
| Learning Rate Scheduler | Cosine Scheduler with Warmup |
| Warmup Ratio | 0.1 |

Table 9: Training Configuration for Auto-regressive Models on the Formal Language Tasks

| Speedup Testing Settings | |
| --- | --- |
| GPU | Nvidia RTX 4090 |
| Batch Size | 1 |
| Sequence Length | 2048 |
| Testing Model Configuration | In Table 7 |

Table 10: Setting for the experiments to test the speedup of MDMs under different sampling steps compare to auto-regressive models.

