# OpenReview forum: "Theoretical Benefit and Limitation of Diffusion Language Model"
_NeurIPS.cc/2025/Conference — NeurIPS 2025 poster_

### Official Review · Reviewer_Gs9S · 2025-06-22

**Clarity:** 3
**Significance:** 2
**Originality:** 2
**Rating:** 4
**Confidence:** 3

**Summary:**

The authors find that the utility of MDMs is highly dependent on the evaluation metric used. When assessed by token-level accuracy, MDMs can achieve near-optimal results with a constant number of sampling steps, regardless of the length of the sequence. However, when evaluated based on sequence-level correctness, which is critical for tasks like logical reasoning, the required number of sampling steps for MDMs must increase linearly with the sequence length. This research provides a theoretical framework for understanding when to use MDMs, suggesting they are well-suited for tasks where fluency is paramount, while auto-regressive models are better for tasks demanding high accuracy and logical consistency.

**Questions:**

Does the metrics TER and SER defined in section 4.1 generalized to conditional generation?

**Ethical Concerns:**

["NO or VERY MINOR ethics concerns only"]

**Final Justification:**

I have read authors' rebuttal and will remain my initial evaluation

**Quality:**

3

**Strengths And Weaknesses:**

Strength
1. The authors study an important problem from a theoretical perspective on the inference of MDMs with comparison to auto-regressive LM.
2. The authors provide insights on the strength of parallel sampling in terms of two types of methods as token-level independent metric and sequence-level holistic accuracy metric.
3. The authors carry out empirical evaluation on synthetic tasks (HMM and n-gram) to corroborate the theoretical finding.

Weakness
1. Though the chosen metrics provide good theoretical finding, it is not directly related to performance of real-world task performance. SER should be too harsh on the success of even reasoning tasks as errors on a few tokens may not have large effects on final accuracy.
2. It is hard to achieve a fair comparison of MDM and AR models especially on real datasets due to differences in model structure and training to isolate the effect of inference. As the authors suggest, even “AR” generation of MDM still leads to gap in performance compared to true AR models.

---

> ### Author Rebuttal · Authors · 2025-07-29
>
> Thank you for your thoughtful feedback. We address the raised concerns below:
>
> ---
>
> # Regarding the practical relevance of SER in reasoning tasks
>
> Thanks — this is an excellent point. We intentionally adopt the correctness — or more precisely, the validity — of the entire trajectory as the evaluation metric for sequence-level assessments, rather than relying solely on the final outcome.
>
> It is widely acknowledged that verifying the correctness of a solution to a mathematical problem requires examining all intermediate steps, not just the final answer (as exemplified by the recent IMO proof verification process). The widespread reliance on final-answer evaluation of LLMs is largely due to the inherent difficulty of automatically assessing the full reasoning chain.
>
> In fact, alongside our work, there has been a surge of recent research revealing the limitations of using only final outcomes. For example, [1] explores how agents can be used to verify reasoning paths, while [2] develops methods for identifying mistakes and correcting outputs. Additionally, a recent survey [3] provides a comprehensive framework for evaluating step-by-step reasoning traces.
>
> We believe that the shift from outcome-based to path-based evaluation will happen soon, and our work will become even more impactful.
>
> [1] Derailer-Rerailer: Adaptive Verification for Efficient and Reliable Language Model Reasoning, arxiv 2408.13940 \
> [2] LLMs cannot find reasoning errors, but can correct them! arxiv 2311.08516v2 \
> [3] Evaluating Step-by-step Reasoning Traces: A Survey, arxiv 2502.12289v1
>
>
> # Regarding the comparison between AR and MDMs
>
> We have made every effort towards a fair comparison, and we believe the conclusion is well-justified.
>
> In the first step, we compare open-sourced MDM and AR models in the left subfigure of Figure 1. However, this comparison inherently favors AR models, which were trained by top-tier industry labs with access to high-quality training data and extensive computational resources. Consequently, the superior performance of AR models in this setting may stem from these advantages rather than from fundamental framework-level differences. **We acknowledge and discuss this limitation in Line 163**.
>
> To address this concern, we adopt a more controlled setting. As noted in Line 164, we decode the MDMs autoregressively from left to right, token by token, and use this as an "AR" baseline for comparison. Admittedly, **this setup inherently favors MDM models, as the "AR" decoding is performed directly using the pre-trained MDMs, without any additional training via next-token prediction. Nonetheless, even under this unfavorable comparison for the AR baseline, MDMs fail to achieve both higher accuracy and greater efficiency.**
>
> Given the above discussion, we believe that both results together well support our conclusion: MDMs do not offer a better efficiency–accuracy trade-off compared to autoregressive models.
>
>
> # Regarding the generalization to conditional generation
> Yes. These metrics can be extended to conditional generation settings with minor modifications.
>
> ---
> Thank you once again for your thoughtful feedback. We hope our clarifications have adequately addressed your concerns, and we would be happy to answer any further questions.

---

### Official Review · Reviewer_QaD3 · 2025-06-30

**Clarity:** 3
**Significance:** 2
**Originality:** 3
**Rating:** 4
**Confidence:** 3

**Summary:**

This paper analyzes the theoretical limitations of masked diffusion models (MDMs). It proposes two evaluation metrics: token error rate and sequence error rate, which aim to quantify the uncertainty in token prediction and the correctness of entire generated sequences, respectively. Through theoretical analysis and toy experiments, the paper shows that while MDMs can achieve low perplexity with few sampling steps, minimizing sequence error requires steps that scale with sequence length.

**Questions:**

- How does SER relate to the performance on reasoning tasks?
- It is mentioned that SER is computed directly using Equation (5). How is the summation over $\\mathcal{L}\_q$ performed in practice, given that the cardinality appears intractable?

**Ethical Concerns:**

["NO or VERY MINOR ethics concerns only"]

**Final Justification:**

The rebuttal addressed most of my concerns.

**Limitations:**

yes

**Paper Formatting Concerns:**

N.A.

**Quality:**

2

**Strengths And Weaknesses:**

Strengths:
- The paper analyzes the theoretical limitations of masked diffusion models, which is an important and timely topic in generative language modeling.

Weaknesses:
- The soundness of the proposed evaluation metric appears questionable. SER is defined as one minus the sum of model probabilities over all sequences in the ground-truth support set $\\mathcal{L}\_q$, regardless of their probabilities under the ground-truth distribution. This leads to unintuitive behavior: if all sequences have non-zero probability under the ground-truth distribution, then SER = 0, regardless of the actual distributional alignment. Consider the following two cases:
    - 1) The model distribution has the same support as the ground truth but a different probability assignment.
    - 2) The model distribution is nearly optimal but assigns a small probability $\\delta$ outside the support.
In these cases, SER would be 0 for the first and $\\delta$ for the second, incorrectly favoring the worse model. This raises concerns about whether SER is a reliable or meaningful metric for analyzing language models or their reasoning capabilities.
- The significance of the theorems is also unclear. Theorems 4.2 to 4.4 provide bounds on the required sampling steps. However, these bounds may be of limited practical value since it's known that MDMs can approximate any ground-truth distribution if the number of sampling steps ≥ the sequence length—because the issue of conditional independence disappears when generating one token per step. (I am not an expert on these proofs and acknowledge the possibility of misinterpretation.)
- The experimental design is limited to toy settings up to 4-gram synthetic languages. While this may help illustrate the theoretical claims, it does not adequately reflect the complexity of real natural language, limiting the broader applicability of the results.

Minor:
- Line 279: \citet

---

> ### Author Rebuttal · Authors · 2025-07-30
>
> Thank you for your feedback and comments. We have carefully considered each point you raised and will address them one by one below.
>
> ---
>
> # Regarding the soundness of the evaluation metric SER
>
> Our primary motivation for introducing SER is to provide a metric for evaluating the validity of reasoning chains, particularly in tasks that require logically correct sequences, such as mathematics or code generation. These tasks involve highly structured data, where the set of correct sequences, $\mathcal{L}_q$, typically occupies a very small subset of the overall sequence space. Therefore, we believe our proposed metric — measuring the probability of generating a sequence outside the correct set — is reasonable.
>
> **This design is novel and has received positive feedback from all other reviewers.** For example:
>
> - “Measuring TER and SER on synthetic tasks could be a useful metric for future research on Diffusion LMs” — Reviewer 4VkA
> - “...For sequence-level metrics such as correctness of reasoning or complete answers, diffusion models fail to offer benefits...” — Reviewer ZofZ
> - “The authors provide insights on the strength of parallel sampling in terms of...sequence-level holistic accuracy metrics” — Reviewer Gs9S
>
> **You proposed two scenarios, and we demonstrate below that neither presents a problem for the SER metric:**
>
> - Case 1: The model distribution has the same support as the ground truth but a different probability assignment. In this scenario, SER = 0. For verifiable reasoning tasks, which may admit multiple correct solutions, SER = 0 is the ideal outcome, indicating that the model exclusively generates correct sequences.
>
> - Case 2: The model distribution is nearly optimal but assigns a small probability $\delta$  outside the support set. Here, SER = $\delta$, which accurately quantifies the model's risk of producing an incorrect output with probability $\delta$.
>
> From the perspective of reasoning accuracy, the model in Case 1 is superior because it never fails, whereas the model in Case 2 will fail with probability of $\delta$. SER correctly captures this distinction, favoring the model that is guaranteed to be correct.
>
> # Regarding the significance of the theorems
>
> **You misinterpret the theoretical results of the paper.** Our target is not to establish the universal approximation capability of MDMs, but rather to rigorously investigate the efficiency–accuracy trade-off (the trade-off between approximation error and computational cost) in comparison to auto-regressive (AR) models. We have emphasized this point multiple times throughout the paper, including in the abstract (Lines 18–20), the introduction (Lines 35–37, 41–42, 64–66, 72–74), Section 4 (Lines 245–250, 292–297), and the conclusion (Lines 363–366).
>
> We demonstrate that the number of sampling steps required by MDMs to achieve a given approximation error varies significantly depending on the evaluation metric:
>
> - For TER (refer to line 240~250): Theorem 4.2 proves that an MDM requires only a constant number of sampling steps (independent of sequence length L) to achieve near-optimal TER. In contrast, AR models inherently require $L$ sequential steps. This result implies that for tasks prioritizing textual fluency (as captured by TER), MDMs offer a substantial theoretical efficiency advantage over AR models.
>
> - For SER (refer to line 268~284): Theorem 4.4 proves that to achieve a low SER, the number of sampling steps required for the MDM must scale linearly with the sequence length L, indicating that the number of neural network executions is comparable between MDMs and auto-regressive models. Consequently, for tasks requiring strict reasoning correctness (as measured by SER), MDMs lose their efficiency advantage.
>
> # Regarding experimental design
>
> **We cannot agree that the paper only contains toy settings. Our theory is primarily motivated by experimental evaluations conducted in practical applications. See lines 155-170 for the detailed discussion.** We observed that on reasoning benchmarks (GSM8k and MBPP), MDMs typically require more sampling steps to reach competitive performance to either AR models or the AR-style generation of MDMs, which diminishes their efficiency advantage.
>
> To better understand these observations, we begin with a theoretical investigation. It is important to note that any theoretical result must be grounded in explicit assumptions. To bound the approximation error of MDMs and analyze the number of steps required to reach that bound, it is essential to specify the underlying ground-truth distribution. To this end, we focus on Hidden Markov Models (HMMs) and n-gram languages, which capture key characteristics of natural language in practical settings, and the theoretical findings reveal that MDMs may require a significant number of steps to achieve a good approximation, which is consistent with the experimental observations.
>
>
> # Regarding the relationship between SER and the performance on reasoning tasks
>
> As stated in our response to Weakness 1, SER is intrinsically linked to performance on reasoning tasks. In many reasoning benchmarks, especially for the verifiable reasoning tasks, there are correct reasoning paths and answers. By verifying the correctness of the entire sequence, SER provides a much stricter evaluation than accuracy, which only considers the final answer. A model that consistently achieves a low SER demonstrates a reliable reasoning capability.
>
> # Regarding the calculation of SER
>
> We have explicitly described how TER and SER are computed in practice. As stated in Line 235, “For robust evaluation, all reported TER and SER values are averages over 2000 generated sequences for each experimental setting.” In fact, we have also tried increasing the number of samples to 20000, and the estimated accuracy shows negligible differences compared to the results with 2000 samples, confirming the reliability and stability of our evaluation.
>
>
> ---
>
> Thank you once again for your time and effort. We hope the clarifications and explanations above have adequately addressed your concerns, and we sincerely look forward to receiving your support.

---

> > ### Comment · Reviewer_QaD3 · 2025-08-05
> >
> > Thank you for your rebuttal.
> >
> > Regarding the soundness of the evaluation metric SER: in the paper, SER is defined with respect to a target language $q$, which I assumed assigns a complex distribution over all possible outputs—for example, assigning higher probability to better answers and lower probability to suboptimal ones. In your rebuttal, it seems only a binary interpretation is considered, where an output is either correct or incorrect. If that is the case, then I agree that SER reduces to a matter of set overlap, and the current definition is sound and essentially equivalent to the usual definition of "accuracy." However, I believe this makes SER somewhat limited, as it cannot capture fine-grained differences in outputs—especially in cases where correctness is ambiguous. In other words, the probability assignment (i.e., the full distribution over the support) is what truly matters.
> >
> > > for tasks requiring strict reasoning correctness (as measured by SER), MDMs lose their efficiency advantage.
> >
> > Apologies if I missed something, but is this conclusion based on the observation that both MDM and AR scale linearly with the sequence length $L$? Or is there a specific justification that AR has a lower multiplicative constant?
> >
> > > By verifying the correctness of the entire sequence, SER provides a much stricter evaluation than accuracy, which only considers the final answer.
> >
> > I am a bit confused and could not find the relevant details in Line 235. Could you clarify how SER is practically computed, especially how $\\mathcal{L}\_q$ is obtained? It seems that $\\mathcal{L}\_q$ should be a set that is not directly enumerable. How is "the correctness of the entire sequence" verified in practice, and how does this differ from verifying reasoning correctness based on the final answer?

---

> > > ### Author Response · Authors · 2025-08-05
> > >
> > > Thanks for your quick reply and further engagement. Please find our replies to your new questions below.
> > >
> > >
> > > # Regarding the SER Metric
> > >
> > > We appreciate that our clarification has resolved the misunderstanding regarding the design philosophy of SER. Indeed, SER can be viewed as a form of sequence-level accuracy. Binary correctness (i.e., right or wrong) is crucial in many scenarios—for example, in math problems, if two proofs generated by different models are both correct, they should be awarded the same score. **Therefore, theoretical results under this setting are both valid and meaningful in their own right.**
> > >
> > > Thank you for the excellent suggestion—fine-grained evaluation is indeed necessary. For instance, the lengthy chains of thought (CoT) produced by current LLMs are often verbose. Therefore, in addition to correctness, factors such as conciseness and overall length should also be considered in the evaluation, which is not captured by the current binary correctness setting. We consider our current results a starting point and plan to extend our framework to support more fine-grained evaluation in future work.
> > >
> > > # Regarding the Inefficiency of MDMs compared to AR when the generation step scales linearly with the sequence length
> > >
> > > It is a really good question. **We have informally described the computational costs of MDM and AR models around Lines 272–274. In response to your comment, we recognize the need for a more rigorous presentation to ensure clarity for the reader.**
> > >
> > > Here, we present a simple case illustrating that when both models generate a length-$L$ sequence in $L$ steps, MDMs are less efficient than AR models by an order of $L$.
> > >
> > > Following the mainstream design choices, we consider the standard Transformer architecture. An AR model has a total computational cost of approximately $O(L^2)$, owing to its sequential generation and KV caching. In contrast, each step of an MDM involves a full forward pass over the entire sequence, resulting in a per-step cost of $O(L^2)$ and a total cost of $O(L^3)$ across $L$ steps. This computational cost analysis can be similarly extended to other architectures, such as Mamba, RWKV, and others. The key reason for this significant difference lies in the fact that MDMs operate over the full sequence at every step and do not support KV caching—a major limitation that remains unresolved to this day.

---

> ### Author Response · Authors · 2025-08-05
>
> # Regarding the Practical Computation of SER
>
> **We have now grasped your message. SER is computed using standard Monte Carlo estimation together with a verifier**. Specifically, we first randomly generate a set of sequences using the model, and then apply the verifier to determine their correctness (i.e., whether the sequence belongs to $ L_q $). SER is then estimated based on the proportion of correct sequences.
>
> Using Monte Carlo estimation is widely adopted in LLM evaluation. For example, in real-world tasks such as GSM8K, Math, and AIME, the mainstream evaluation protocol involves sampling outputs using $k$ different random seeds and checking whether at least one correct solution exists—commonly referred to as pass@$k$, where $k$ is usually a small number.
>
> Developing the verifier is non-trivial. In our experimental setting for synthetic tasks, the ground-truth language $\mathcal{L}_q$ (an HMM with sparse transitions) is formally defined (construction details in Appendix F.1). We implement a perfect verifier that deterministically checks whether a generated sequence contains invalid transitions using the forward algorithm [1,2], thereby verifying whether the sequence belongs to $ \mathcal{L}_q $. The reported TER and SER values are computed by averaging the verification outcomes over 2,000 generated sequences per experimental setting. We further verified the stability of this estimate using 20,000 samples, which showed negligible differences.
>
> For practical reasoning tasks, various verification methods can be employed, such as final-answer checkers or neural network-based verifiers. Each verifier provides a different surrogate for SER. For instance, if correctness is judged solely based on the final answer, the resulting metric serves as a lower bound on the true SER—since a correct final answer may still contain errors in intermediate steps. By incorporating stronger verifiers that evaluate the full reasoning chain, the estimated SER can more closely approximate the true value.
>
> **We would like to thank you for highlighting this important writing issue twice. At a minimum, we should have clearly presented the general workflow for estimating SER, as well as the specific algorithm used in our experiments. Since NeurIPS does not allow direct revision of the manuscript, we would like to detail the intended modifications here:**
>
> a. In Section 4.1, we will introduce the general methodology for empirically estimating SER using a verifier combined with Monte Carlo sampling.
>
> b. In Appendix F.1, we will provide a detailed description of the SER computation procedure for the target HMM experiment, including the construction of the verifier.
>
> Thank you again for your questions and valuable comments, which have helped us identify key issues in our presentation. We hope this clarifies your concerns, and we greatly appreciate your insightful feedback.
>
> [1] Wikipedia contributors, "Forward algorithm," Wikipedia, The Free Encyclopedia, https://en.wikipedia.org/wiki/Forward_algorithm (accessed August 5, 2025).
> [2] Eddy, Sean R. "Hidden markov models." Current opinion in structural biology 6, no. 3 (1996): 361-365.

---

> > ### Comment · Reviewer_QaD3 · 2025-08-08
> >
> > Thanks for the clarification on the Monte Carlo estimation of SER. This makes sense to me.
> >
> > Regarding SER, I still have some questions. Based on your description, I am still unclear about the real difference between SER and accuracy. As you mentioned in your previous post, "SER provides a much stricter evaluation than accuracy," I assume that when evaluating SER, it is not only the correctness of the final answer that is considered. How was this specifically implemented in your experiments?

---

> ### Author Response · Authors · 2025-08-08
>
> Thank you for your follow-up question. **Your understanding is correct--when evaluating SER, it is not only the correctness of the final answer that is considered.** While accuracy considers only the final outcome, SER assesses the validity of the entire reasoning process. For example, consider the solution path to a math problem: $1 + 2 + 3 = 4 + 3 = 6$. As the final answer is correct, accuracy would label it as correct. But SER would mark it as incorrect due to errors in the intermediate steps.
>
> **We are also somewhat puzzled by your repeated question regarding how SER was specifically implemented in our experiments.** If you are referring to the experiment in Section 5, we have already described the implementation in our previous response. Nevertheless, we would be happy to clarify it once again here.
>
> In the experiment in Section 5, we study a synthetic setting in which sequences are generated by a sparse HMM model. After training the neural network model, we generate a set of sample sequences. To calculate SER, we determine whether each generated sequence has non-zero probability under the ground truth sparse HMM—this is equivalent to checking whether the sequence contains any invalid transitions with respect to the sparse HMM. This evaluation can be efficiently performed using the forward algorithm [1][2].
>
> [1] Wikipedia contributors, "Forward algorithm," Wikipedia, The Free Encyclopedia, https://en.wikipedia.org/wiki/Forward_algorithm (accessed August 5, 2025).
> [2] Eddy, Sean R. "Hidden markov models." Current opinion in structural biology 6, no. 3 (1996): 361-365.
>
> In the previous response, we also described how SER can be evaluated in general tasks through the development of strong verifiers, and we plan to make sufficient modifications to the paper regarding this clarification.
>
> **If you are referring to the experiment in Figure 1, please note that it serves as a motivating example and is not related to the computation or evaluation of SER.** The reported accuracy only concerns the correctness of the final answer, which, as we discussed in our previous response, represents an upper bound on $1-\text{SER}$.
>
> We hope this explanation addresses your concerns. Thank you again for your valuable feedback, and we kindly ask you to reconsider your evaluation.

---

> > ### Comment · Reviewer_QaD3 · 2025-08-08
> >
> > Thanks for your further clarification.
> > > when evaluating SER, it is not only the correctness of the final answer that is considered. While accuracy considers only the final outcome, SER assesses the validity of the entire reasoning process. For example, consider the solution path to a math problem: $1 + 2 + 3 = 4 + 3 = 6$. As the final answer is correct, accuracy would label it as correct. But SER would mark it as incorrect due to errors in the intermediate steps.
> >
> > My question was specifically about such examples. In evaluating SER for these reasoning cases, how is an intermediate wrong step like "$4 + 3$"recognized? Is it because, under the HMM setting, this is straightforward since the likelihood of the entire path can be evaluated? For more general natural language cases, estimating SER seems challenging, and in such cases would we have to fall back to a degraded setting where SER essentially characterizes the same thing as accuracy? In other words, for general natural language tasks, it is difficult to automatically determine whether an output $x\\in\\mathcal{L\_q}$ , and thus difficult to obtain more information than accuracy alone. This is where I still have concerns about whether such a new metric actually brings additional value.

---

> ### Author Response · Authors · 2025-08-08
>
> Thank you for your engagement. We’re glad to have addressed most of your concerns and now arrive at the final—-and perhaps most critical—-one: the significance of the SER metric. To address this, we present a three-stage response.
>
> ## Regarding whether "SER essentially characterizes the same thing as accuracy"
>
> The potential mismatch between the correctness of the final output and that of the reasoning chain is a well-recognized issue and remains an active area of research in recent studies on large language models (LLMs). For instance, [1][2][3] demonstrate that when solving math problems, a model may apply an incorrect formula or make multiple logical errors that coincidentally cancel out—still yielding the correct final answer. Similarly, [4][5][6] show that a model might guess the correct answer without understanding how to derive it, generating a completely fallacious reasoning process.
>
> **Based on these related works, the intuition--"SER essentially characterizes the same thing as accuracy"-- is not correct.**
>
>    [1] Solving Quantitative Reasoning Problems with Language Models, arxiv 2206.14858.
>    [2] Unfaithful explanations in chain-of-thought prompting, arxiv 2305.04388.
>    [3] Chain-of-thought reasoning in the wild is not always faithful, arxiv 2503.08679.
>    [4] Language Models Don't Always Say What They Think: Unfaithful Explanations in Chain-of-Thought Prompting, arxiv 2305.04388.
>    [5] Measuring Faithfulness in Chain-of-Thought Reasoning, arxiv 2307.13702.
>    [6] Measuring the Faithfulness of Thinking Drafts in Large Reasoning Models, arxiv 2505.13774.
>
>
> ## Regarding the essentials of the developing process-level metric
>
> **As SER doesn't equal final accuracy, we next demonstrate that its development is essential.** Reviewer Gs9S raised a similar question, and we would like to quote our response to Gs9S here.
>
> It is widely acknowledged that verifying the correctness of a solution to a mathematical problem requires examining all intermediate steps, not just the final answer (as exemplified by the recent IMO proof verification process). The widespread reliance on final-answer evaluation of LLMs is largely due to the inherent difficulty of automatically assessing the full reasoning chain.
>
> In fact, alongside our work, there has been a surge of recent research revealing the limitations of using only final outcomes. For example, [7] explores how agents can be used to verify reasoning paths, while [8] develops methods for identifying mistakes and correcting outputs. Additionally, a recent survey [9] provides a comprehensive framework for evaluating step-by-step reasoning traces.
>
> **Although it is challenging, we believe that the shift from outcome-based to path-based evaluation will happen soon, and our work will become even more impactful.**
>
> [7] Derailer-Rerailer: Adaptive Verification for Efficient and Reliable Language Model Reasoning, arxiv 2408.13940.
> [8] LLMs cannot find reasoning errors, but can correct them! arxiv 2311.08516v2.
> [9] Evaluating Step-by-step Reasoning Traces: A Survey, arxiv 2502.12289v1.
>
> ## Regarding the computation of SER with a verifier.
>
> Regarding the computation of SER. As mentioned in our previous responses, with a sufficiently powerful verifier, we can use the Monte Carlo method to calculate SER. The accuracy of the SER calculation is directly linked to the capability of the verifier; a more powerful verifier yields a more precise result. **[10] shows that a strong verifier can help Gemini find the correct reasoning trajectory and solve IMO-level problems, and your concern about detecting $1+2+3=4+3$ should be easily achieved.**
>
> [10] Gemini 2.5 Pro Capable of Winning Gold at IMO 2025 https://arxiv.org/pdf/2507.15855
>
> **Once again, we sincerely thank you for raising these thoughtful questions. They have helped us clarify our work, and we would be happy to incorporate these discussions into the paper if you believe they would be beneficial to readers.**

---

### Official Review · Reviewer_ZofZ · 2025-07-02

**Clarity:** 3
**Significance:** 3
**Originality:** 4
**Rating:** 5
**Confidence:** 4

**Summary:**

This work theoretical studies the benefits and limitations of Masked diffusion models for text generation. It first empirically shows that existing autoregressive models such as qwen perform better in terms of efficiency and accuracy than diffusion models such as Llada, which claims better efficiency. It is shown that for n-gram langauges diffusion models can outperform autoregressive models in efficiency with competitive accuracy at token-level metrics such as perplexity. However, when it comes to sequence-level tasks such as correctness of answers or logic, it is shown that there exists an Hidden Markov Model language such that diffusion models show no such advantage over autoregressive models.

Experimental results on synthetic data shows similar pattern as claimed by the theorems, i.e., for sequence-level metrics, the performance of diffusion models are much worse than autoregressive models. However, for token-level metrics the performance of diffusion model looks nearly independent of the number of steps taken by the diffusion model, and the performance of diffusion and autoregressive models look comparable.

**Questions:**

Please also answer the weaknesses.

Additional Questions
-  Line 248: even though it is claimed that the number of samples required is constant. Is it not dependent on the Markovity n of the language model?

- Why is the TER result in Thm. 4.2 only for n-gram language whereas the SER result in Thm. 4 gives an existence result but does not mention the n-gram of the language. Does the result in Thm. 4 claim the existence of an n-gram language for which the theorem holds or is it for any possible Markovian language?

- Since diffusion models perform well on token-level metric but not on sequence level metric. Can we not modify the sampling algorithm for using speculative decoding? That is, use diffusion model as a draft model and use autoregressive to fix sequence level metric. This is a general question. No need for theoretical analysis or experiments. I am trying to understand if the limitation of diffusion models shown a major issue for sequence-level metric or is there a way around that is not discussed.

**Ethical Concerns:**

["NO or VERY MINOR ethics concerns only"]

**Final Justification:**

I have read through the authors' rebuttal as well as other reviews and their rebuttals.

The authors' rebuttal addresses my concerns, especially, adding experiments with other models such as Llama, as well as pointing out other experiments in the appendix.

Overall, I find the results showing that MDMs are much worse than Autoregressive Models in tasks requiring sequence correctness very interesting and useful for the LLM reasoning community.

Hence, I update my score to accept

**Limitations:**

yes

**Paper Formatting Concerns:**

No concerns

**Quality:**

4

**Strengths And Weaknesses:**

Strengths
- The paper provides interesting theoretical results showing that the performance of emerging diffusion language models are dependent on the nature of the evaluation metrics, in particular, for token level metrics diffusion models show promise of outperforming autoregressive models. But for sequence-level metrics such as correctness of reasoning or complete answer, diffusion model fails to offer benefits compared to autoregressive models. These results are shown for Markov-style language assumption, nevertheless, the results are interesting.

- Experimental results on synthetic data validates both the claims for token-level as well as sequence-level metrics.


Weaknesses
- One weakness already mentioned in the paper is that all theoretical analysis and experimental results are restricted to HMM-style language. However, a) this assumption may not hold for general tasks, b) even if the assumption of HMM holds, tasks like infilling may benefit from diffusion models more. Even if the theoretical analysis for such non-HMM languages may be difficult, the paper must have some experimental evaluations provided to better understand the possible generalization of the results in the paper.

- Fig 1 compares performance of qwen versus Llada models on GSM8k to claim the poor performance of diffusion models compared to autoregressive models. But to better understand the result, we should compare with more available models of comparable sizes such as llama 8b and also test on various other benchmarks such as Math500, HumanEval, MBPP, etc.

---

> ### Author Rebuttal · Authors · 2025-07-30
>
> We sincerely thank you for your insightful feedback and suggestions. Below, we address the listed weaknesses and questions in detail.
>
> ---
> # Regarding the use of HMMs and experimental evaluations
>
> **Our theory is primarily motivated by experimental evaluations conducted in practical applications**, i.e., on reasoning benchmarks (GSM8k and MBPP), we observed that MDMs typically require more sampling steps to reach competitive performance to either AR models or the AR-style generation of MDMs, which diminishes their efficiency advantage.
>
> To better understand these observations, we begin with a theoretical investigation. It is important to note that any theoretical result must be grounded in explicit assumptions. To bound the approximation error of MDMs and analyze the number of steps required to reach that bound, it is essential to specify the underlying ground-truth distribution. To this end, we focus on Hidden Markov Models (HMMs) and n-gram languages, which capture key characteristics of natural language in practical settings, and the theoretical findings reveal that MDMs may require a significant number of steps to achieve a good approximation, which is consistent with the experimental observations.
>
> As you rightly point out, certain gaps between theory and practice still remain.  Moving forward, we plan to follow your suggestion and extend our analysis to more realistic scenarios, including long-range dependency structures, syntactic hierarchies, and distributions derived from large-scale language model pretraining corpora.
>
> # Regarding infilling tasks
>
> Specialized tasks, such as bidirectional infilling, can be effectively addressed using dedicated frameworks like BERT (masked language modeling) and diffusion models, which take the entire sequence as input and generate missing tokens conditioned on the surrounding context. In this paper, however, we focus on general NLP tasks. We agree that your suggestion — to compare MDM and AR on more specific tasks — is insightful, and we consider it a valuable direction for future work.
>
> # Regarding broader model comparison and benchmarks
>
> In the submitted version, we have already included experiments on MBPP in addition to GSM8K. The observations and conclusions are consistent across both benchmarks. Due to space limitations, the MBPP results are presented in Appendix A.3, as referenced in Line 160.
>
> To further address your concern, during the rebuttal period, we extended our comparison to include LLaMA 8B and conducted additional experiments on the HumanEval benchmark. The results are shown in the table below （* indicates AR baselines）. The conclusions remain consistent: MDMs fail to achieve both higher efficiency and better performance simultaneously in realistic settings.
>
> | Model                           | Accuracy | Time cost (s) |
> | :------------------------------ | -------- | ------------- |
> | Dream-v0-Base-7B, step 128      | 15.9     | 499.5         |
> | Dream-v0-Base-7B, step 256      | 32.9     | 954.6         |
> | Dream-v0-Base-7B, step 512      | 56.1     | 1838.7        |
> | Dream-v0-Base-7B, left-to-right | 57.9*     | 784.1*         |
> | LLaDA-8B-Base, step 128         | 12.8     | 626.2         |
> | LLaDA-8B-Base, step 256         | 26.8     | 1143.9        |
> | LLaDA-8B-Base, step 512         | 32.9     | 2058.9        |
> | LLaDA-8B-Base, left-to-right    | 28.7*     | 1842.4*        |
> | Qwen2.5-7B                      | 56.7*     | 145.5*         |
> | Llama-3-8B                      | 62.2*     | 147.0*         |
>
>
> # Regarding the dependency of the number of steps
>
> Thanks for the question. The number of steps in the theorem depends on the order $n$ of the Markov model (See line 238). The "constant" here mainly refers to the dependency on sequence length. We will make it clear in the next version.
>
> # Regarding the HMM in Theorem 4.4
>
> Good catch — HMMs and n-gram language models are closely related. It's straightforward to show that all n-gram languages can be represented as HMMs, and HMMs can be transformed into n-gram models under certain conditions (e.g., finite state and finite-memory assumptions; a direct proof can be obtained using GPT).
>
> The SER result in Theorem 4.4 also holds for n-gram languages, as the construction used in the proof of Theorem 4.4 can be viewed as an n-gram language with some modifications (detailed in Appendix E.7 and E.8). However, using the HMM framework offers several advantages, particularly in simplifying the proof construction and making it easy for the reader to understand. We will carefully consider both the unification of the proof and the consistency of the overall framework.
>
> # Regarding combining AR and MDM
>
> This is an excellent suggestion — integrating diffusion-based text generation with speculative decoding is a very promising direction.
>
> ---
>
> Thank you once again for your time and effort. We hope the clarifications and explanations above have adequately addressed your concerns.

---

### Official Review · Reviewer_4VkA · 2025-07-04

**Clarity:** 4
**Significance:** 3
**Originality:** 3
**Rating:** 5
**Confidence:** 3

**Summary:**

This paper provides an analysis of the properties of MDMs in the context of language modeling and compares them to autoregressive LMs. According to the paper's findings, the number of steps required to achieve near-optimal perplexity or a reasonable Token Error Rate (TER) for MDMs is relatively constant, regardless of sequence length. Conversely, the number of steps needed to achieve a specific Sequence Error Rate (SER) scales linearly with increasing input sequence length. Theoretical findings are validated on synthetic tasks, and the experimental results confirm these theoretical insights.

**Questions:**

Q1: Does Theorem 4.4 hold for continuous diffusion language models like [1]?
Q2: Diffusion models are less sample-efficient during training, as each training step involves only a single noise level [2]. Would the results of the experiments in Section 5.1 change if the number of training epochs for MDMs were increased compared to the AR model?
Q3: Where would be Qwen 2.5 3B and Qwen 2.5 1.5B in Figure 1?

[1] [Likelihood-Based Diffusion Language Models](https://arxiv.org/pdf/2305.18619)
[2] https://sander.ai/2023/01/09/diffusion-language.html

**Ethical Concerns:**

["NO or VERY MINOR ethics concerns only"]

**Final Justification:**

During the rebuttal phase, the authors provided additional experiments on the HumanEval benchmark, which addressed my earlier concerns regarding the limited scope of their evaluation. They also included additional baselines using smaller autoregressive models, further strengthening their empirical analysis.

**Limitations:**

The authors adequately addressed the limitations

**Paper Formatting Concerns:**

I didn't notice any formatting issues.

**Quality:**

3

**Strengths And Weaknesses:**

Strengths

 - S1: This contribution could be very important for the field of diffusion language models. Diffusion LMs are believed to be more efficient on code and math, however, this work suggests that the number of steps scales linearly with sequence length to achieve low SER. Therefore, using diffusion for this task could be less efficient compared to AR models.
 - S2: Measuring TER and SER on synthetic tasks could be a useful metric for future research on Diffusion LMs.
 - S3: The paper is easy to follow because of the structure.

Weaknesses

- W1: The main text only includes results on the GSM8K dataset. Adding additional datasets and reporting mean accuracy across them would help address this limitation.
 - W2: The apparent superiority of the AR model in Figure 1 might be explained by differences in the dataset used and the amount of FLOPs during training.
 - W3: Setup where #steps < sequence length should be compared to AR model with lower number of parameters during inference, matching the speed or computational cost (FLOPs) of MDMs like LLaDA with lower number of steps.

---

> ### Author Rebuttal · Authors · 2025-07-30
>
> Thank you for your valuable feedback. Here, we address your concerns and questions below:
>
> ---
>
> # Regarding experiments on additional benchmarks
>
> **In the submitted version, we have already included experiments on MBPP in addition to GSM8K.** The observations and conclusions are consistent across both benchmarks. Due to space limitations, the MBPP results are presented in Appendix A.3, as referenced in Line 160.
>
> To further address your concern, during the rebuttal period, we extended our comparison to include LLaMA 8B, Qwen2.5-3B and Qwen2.5-1.5B, and conducted additional experiments on the HumanEval benchmark. The results are shown in the table below (* indicates AR baselines). The conclusions remain consistent: MDMs fail to achieve both higher efficiency and better performance simultaneously in realistic settings.
>
> | Model                           | Accuracy on HumanEval | Time cost (s) |
> | :------------------------------ | -------- | ------------- |
> | Dream-v0-Base-7B, step 128      | 15.9     | 499.5         |
> | Dream-v0-Base-7B, step 256      | 32.9     | 954.6         |
> | Dream-v0-Base-7B, step 512      | 56.1     | 1838.7        |
> | Dream-v0-Base-7B, left-to-right | 57.9*     | 784.1*         |
> | LLaDA-8B-Base, step 128         | 12.8     | 626.2         |
> | LLaDA-8B-Base, step 256         | 26.8     | 1143.9        |
> | LLaDA-8B-Base, step 512         | 32.9     | 2058.9        |
> | LLaDA-8B-Base, left-to-right    | 28.7*     | 1842.4*        |
> | Qwen2.5-7B                      | 56.7*     | 145.5*         |
> | Qwen2.5-3B                      | 39.0*     | 244.5*         |
> | Qwen2.5-1.5B                    | 37.2*     | 203.1*         |
> | Llama-3-8B                      | 62.2*     | 147.0*         |
>
>
> # Regarding the performance gap between AR and MDMs
>
> **We have made every effort towards a fair comparison, and we believe the conclusion is well-justified.**
>
> In the first step, we compare open-sourced MDM and AR models in the left subfigure of Figure 1. As you point out, this comparison inherently favors AR models, which were trained by top-tier industry labs with access to high-quality training data and extensive computational resources. We acknowledge and discuss this limitation in Line 163.
>
> To address this concern, we have already adopted a more controlled setting. As noted in Line 164, we decode the MDMs auto-regressively from left to right, token by token, and use this as an "AR" baseline for comparison. Admittedly, this setup inherently favors MDM models, as the "AR" decoding is performed directly using the pre-trained MDMs, without any additional training via next-token prediction. Nonetheless, even under this unfavorable comparison for the AR baseline, MDMs fail to achieve both higher accuracy and greater efficiency.
>
> Given the above discussion, we believe that both results together well support our conclusion: MDMs do not offer a better efficiency–accuracy trade-off compared to auto-regressive models.
>
> # Regarding comparison to smaller AR models
>
> To address your concerns that MDMs with less steps should be compared with AR models with lower number of parameters, we conducted experiments to test the performance of Qwen2.5-3B and Qwen2.5-1.5B on GSM8K, MBPP and HumanEval benchmarks. The results on the HumanEval benchmark are listed in the above table, and results on GSM8K and MBPP are shown below. It is clear that in order to achieve comparable accuracy on the benchmarks, MDMs require much more time during inference, which aligns with our conclusion that MDMs do not offer a better accuracy-efficiency trade-off. For potential concerns on the fairness of comparison, please refer to the previous section.
>
> | Model                           | Accuracy on GSM8K | Time cost (s) |
> | :------------------------------ | -------- | ------------- |
> | Dream-v0-Base-7B, step 64      | 47.5     | 1349.9         |
> | Dream-v0-Base-7B, step 128      | 68.6     | 2649.5         |
> | Dream-v0-Base-7B, step 256      | 76.3     | 5253.7         |
> | LLaDA-8B-Base, step 64         | 54.5     | 2803.3         |
> | LLaDA-8B-Base, step 128         | 66.0     | 5413.7         |
> | LLaDA-8B-Base, step 256         | 69.1     | 10696        |
> | Qwen2.5-3B                      | 76.5*     | 553.3*         |
> | Qwen2.5-1.5B                    | 63.2*     | 448.6*         |
>
> | Model                           | Accuracy on MBPP | Time cost (s) |
> | :------------------------------ | -------- | ------------- |
> | Dream-v0-Base-7B, step 128      | 26.4     | 2380.1         |
> | Dream-v0-Base-7B, step 256      | 42.8     | 4677.0         |
> | Dream-v0-Base-7B, step 512      | 56.6     | 9197.0
> | LLaDA-8B-Base, step 512         | 39.6     | 10839        |
> | Qwen2.5-3B                      | 55.2*     | 213.2*         |
> | Qwen2.5-1.5B                    | 43.4*     | 166.2*         |
>
> # Regarding continuous diffusion language models
>
> No. Theorem 4.4 is specialized for the discrete masked diffusion model. It cannot be extended to continuous diffusion language models [1].
>
> [1] Likelihood-Based Diffusion Language Models, arxiv 2305.18619
>
> # Regarding experiments in Section 5.1
>
> In our experiments, both MDMs and AR models were trained until convergence. While the optimal perplexity in these synthetic datasets can be calculated, our trained MDMs and AR models have achieved the optimal perplexity on the validation sets (refer to line 316). We believe the comparison is fair enough.
>
> ---
>
> Thank you again for your thoughtful feedback. We hope that these clarifications and additions will enhance the strength and clarity of the paper’s contributions, and we would be happy to answer any further questions.

---

> > ### Comment · Reviewer_4VkA · 2025-08-02
> >
> > I thank the authors for their detailed response.
> >
> > **Experiments on Additional Benchmarks**
> >
> > > In the submitted version, we have already included experiments on MBPP in addition to GSM8K.
> >
> > I can see GSM8K in the main text (Figure 1) and MBPP in the Appendix (Figure 2). This is what I was referring to.
> >
> > Thank you for including the additional results -- I believe this resolves my concern regarding benchmark coverage.
> >
> > **Other Points**
> >
> > Thank you for the clarifications and for adding the smaller AR models.
> >
> > ----
> > Your response addresses my concerns, and I will adjust my score accordingly.

---

> > > ### Author Response · Authors · 2025-08-03
> > >
> > > Thank you for acknowledging our rebuttal and adjusting the score. If there are any further questions or suggestions during the subsequent process, please feel free to let us know. Thank you again for your time and effort!

---

### Note · Authors · 2025-08-14

**We thank AC for handling this work and thank all reviewers for their constructive and thoughtful evaluation of our work. All agree that our study is both interesting and important to generative language modeling, particularly in advancing the understanding of diffusion-based models.** Through multiple rounds of discussion—especially with Reviewer QaD3, who engaged in depth—we believe we have clarified all key points and fully addressed the concerns raised.

The reviewers provided several valuable suggestions that helped us identify presentation issues to address in the next version, including:

**Regarding the extended experiments.**

Reviewers 4VkA, ZofZ, and Gs9S requested additional empirical analysis. We explained that, **due to length constraints, some results were already included in the appendix.** We have since conducted further experiments to address their concerns and plan to integrate these results into the extended version of the paper.

**Regarding the practical relevance of sequence error rate (SER).**

Traditional benchmarks focus primarily on final-answer accuracy. Reviewer QaD3 and Gs9S have concerns about whether developing a sequence error rate is helpful. In fact, our choice of using SER anticipates the growing shift toward path-based evaluation, supported by recent literature.

In addition, Reviewer QaD3 inquired about the implementation of our method for calculating SER. We clarified the distinction between SER and final-output accuracy, described our implementation in the synthetic experiment, and explained how SER could be computed in practical tasks using a verifier and Monte Carlo sampling—the standard approach for solving complex mathematical problems by LLMs, as exemplified in IMO. These discussions were valuable, and we plan to incorporate them into the paper for greater clarity.

----

We greatly appreciate the reviewers’ responsibility in engaging with our work, which has helped strengthen the presentation and framing of our work. We hope this remark clarifies our key contributions and effectively concludes the discussions.

---

### Decision · Program_Chairs · 2025-09-17

**Decision:**

Accept (poster)

**Comment:**

This paper investigates the theoretical benefits and limitations of diffusion LLMs. By modeling natural language generation through n-grams or hidden Markov models, the authors derive theoretical bounds on both token-level and sentence-level error rates. A key finding is that, in the worst case, the number of required sampling steps must scale linearly with sequence length in order to control the sequence-level error rate—thereby questioning the efficiency advantages often attributed to diffusion LLMs. While reviewers initially raised concerns about the limited experimentation with state-of-the-art diffusion language models, these were satisfactorily addressed during the rebuttal discussion. In light of the overall positive reviews, I recommend acceptance.